# Cooperative microbial interactions drive spatial segregation in porous environments

Yichao Wu[1,12], Chengxia Fu[1,12], Caroline L. Peacock [2], Søren J. Sørensen [3], Marc A. Redmile-Gordon[4], Ke-Qing Xiao [2,5], Chunhui Gao [1], Jun Liu[1], Qiaoyun Huang [1], Zixue Li[6], Peiyi Song[6], Yongguan Zhu [5,7], Jizhong Zhou [8,9,10,11] & Peng Cai [1] ✉

The role of microbial interactions and the underlying mechanisms that shape complex biofilm communities are poorly understood. Here we employ a microfluidic chip to represent porous subsurface environments and show that cooperative microbial interactions between free-living and biofilm-forming bacteria trigger active spatial segregation to promote their respective dominance in segregated microhabitats. During initial colonization, free-living and biofilm-forming microbes are segregated from the mixed planktonic inoculum to occupy the ambient fluid and grain surface. Contrary to spatial exclusion through competition, the active spatial segregation is induced by cooperative interactions which improves the fitness of both biofilm and planktonic populations. We further show that free-living *Arthrobacter* induces the surface colonization by scavenging the biofilm inhibitor, D-amino acids and receives benefits from the public goods secreted by the biofilm-forming strains. Collectively, our results reveal how cooperative microbial interactions may contribute to microbial coexistence in segregated microhabitats and drive subsurface biofilm community succession.

The terrestrial and oceanic subsurface hosts over 80% of microorganisms on Earth and is thus the major microbial habitat on our planet[1,2]. Unlike aqueous environments (e.g. open ocean) where microbes are mostly free living (planktonic), the subsurface provides immensely large surface area for microbial attachment. The surface-attached microbes sequester nutrients from pore water and grow into dense multi-species assemblages, called biofilms[3,4]. The close proximity of diverse species in biofilms facilitates various interactions between them, such as quorum sensing and synergistic metabolism, which determine the community traits and functions[5–7].

In the past decades, theoretical and experimental research has been performed to dissect the intricate interactions that dictate subsurface biofilm community structure. Cooperative microorganisms such as cross-feeding partners are found to co-aggregate in biofilm communities to allow reciprocal benefits[8–10]. By contrast, the mutually antagonistic microorganisms tend to exclude each other from local niches and segregate spatially[7,11]. Besides direct functional

[1]State Key Laboratory of Agricultural Microbiology, College of Resources and Environment, Huazhong Agricultural University, Wuhan, China. [2]School of Earth and Environment, University of Leeds, Leeds LS2 9JT, UK. [3]Section of Microbiology, Department of Biology, University of Copenhagen, Copenhagen, Denmark. [4]Department of Environmental Horticulture, Royal Horticultural Society, Wisley Surrey GU23 6QB, UK. [5]State Key Laboratory of Urban and Regional Ecology, Research Center for Eco-Environmental Sciences, Chinese Academy of Sciences, Beijing, China. [6]School of Physics, Huazhong University of Science and Technology, Wuhan, China. [7]Key Laboratory of Urban Environment and Health, Institute of Urban Environment, Chinese Academy of Sciences, Xiamen, China. [8]Institute for Environmental Genomics and Department of Microbiology and Plant Biology, University of Oklahoma, Norman, USA. [9]State Key Joint Laboratory of Environment Simulation and Pollution Control, School of Environment, Tsinghua University, Beijing, China. [10]Earth and Environmental Sciences, Lawrence Berkeley National Laboratory, Berkeley, USA. [11]School of Civil Engineering and Environmental Sciences, University of Oklahoma, Norman, USA. [12]These authors contributed equally: Yichao Wu, Chengxia Fu. ✉e-mail: cp@mail.hzau.edu.cn

consequences, the physical structure of subsurface environments can determine ecological stability and functional activities by modulating the spatial distribution of cooperative and competitive genotypes. Compared with well-mixed environments, spatial segregation under structured conditions balances the competitive and cooperative interactions to stabilize the community[12]. For instance, physical separation in porous media enables the coexistence of slow-growing species with fast-growing competitors, as the rapid biofilm formation blocks fluid flow and redirects nutrients to its competitors[13,14]. One recent experiment also demonstrated that the spatial segregation of biofilm consortia governs the metabolite cross-feeding and microbial growth via tuning the fidelity of quorum sensing signal transmission[15]. The current understanding of interaction-derived subsurface biofilm communities however, largely rests on dual-species communities. How microbial interactions shape diverse biofilm communities in spatially structured environments is still poorly understood.

Here, we investigated the biofilm colonization process in a porous medium where soil bacteria self-assemble into structured microbial communities. Using microfluidics, 16S rRNA gene amplicon sequencing, and fluorescence in situ hybridization (FISH), we observed that during early biofilm development, biofilm-deficient species actively primed the microscale environment for biofilm-forming microbes to colonize surfaces. We further performed exometabolomics,

transcriptomics, pairwise interaction analyses and genetic manipulation to uncover the mechanisms of interspecific interactions. We find that the interaction between biofilm-deficient and biofilm-forming species drives microbial community succession through active spatial segregation in the subsurface environment.

## Results

### Biofilm development in the porous medium

We developed a microfluidic chip consisting of an array of pillars (analogous to subsurface grains) to resemble the subsurface porous media (Fig. 1a and Supplementary Fig. 1). The colloidal transport and retention in the microfluidic chamber were characterized to evaluate the residence of planktonic cells. Based on the breakthrough curve of fluorescent microspheres (Supplementary Fig. 2), the mean travel time of colloid particles was $51.48 \pm 2.16$ min. It suggests that both biofilm and planktonic cells have sufficient residence time to grow in the porous medium. The microfluidic device was inoculated with a complex microbial community extracted from soil and then supplied with the soil extract medium prepared from the same soil sample. Biofilm development was initiated with small microbial colonies formed on the grain (Fig. 1b). After 72 h growth, biofilms started to blanket the grain surface and extended into the pore space and eventually clogged the channels of the porous matrix.

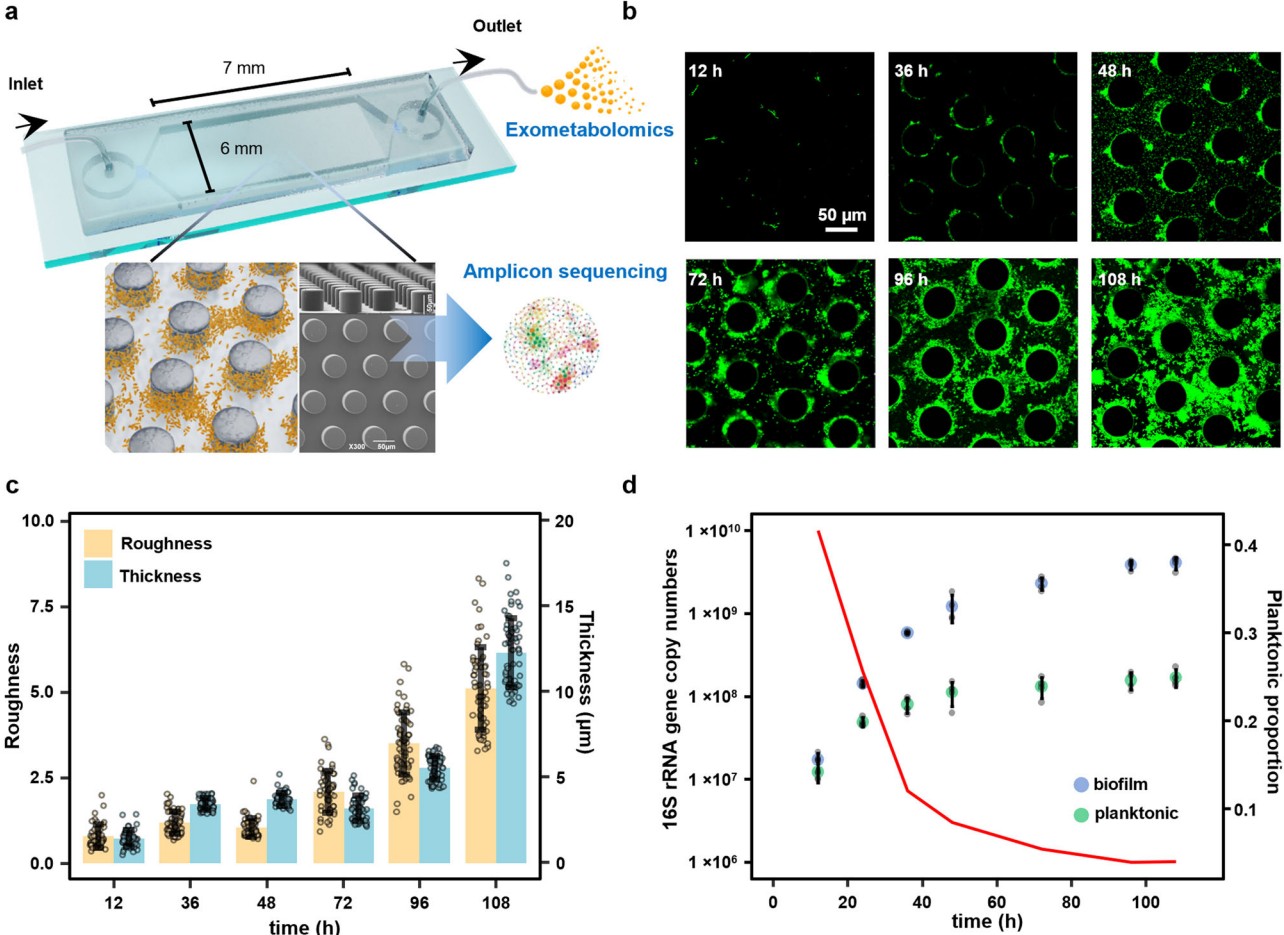

**Fig. 1 | Biofilm development in the microfluidic chip. a** Schematic of the microfluidic device that mimics the subsurface porous environment. Microbial community composition is profiled via 16S rRNA gene amplicon sequencing. The effluent is subject to exometabolomic analysis. **b** The development of biofilm architecture in the porous environment. Biofilm cells are stained with SYTO 9 (green). Scale bar represents 50 μm. The experiment was repeated in four independent microfluidic chips with similar results. **c** The dynamics of biofilm thickness

and roughness ($n = 4$ chips × 15 grains). Biofilm roughness was calculated as the standard deviation of biofilm thickness on individual grain surfaces. Data are presented as mean values ± standard deviation. **d** The amount of planktonic and biofilm cells in the microfluidic chamber determined by qPCR ($n = 4$ chips). The solid line shows a decreasing proportion of planktonic cells with increasing biofilm development. Data are presented as mean values ± standard deviation.

Based on quantitative imaging analysis, biofilm roughness increased significantly after 72 h (Fig. 1c). The increased roughness, typically indicating the formation of mature biofilms, enlarges the liquid-biofilm interface to allow efficient mass transfer of nutrients to biofilms[16,17]. The amount of planktonic and biofilm cells was quantified via qPCR. Both the growth profiles of planktonic and biofilm cells in the microfluidic chamber followed a logistic growth pattern, which reached a plateau at the end of the incubation (Fig. 1d). Except for the initial 24 h, the amount of biofilm cells was considerably higher than that of the planktonic community.

## Succession of microbial community in porous medium

To investigate the community succession during biofilm development, the total microbial community dynamics in the microfluidic chips were monitored by 16S amplicon sequencing. The Bray-Curtis dissimilarity of total microbial communities between adjacent time intervals decreased as biofilm growth progressed (Fig. 2a). During the initial stage, the community diversity and richness exhibited a significant decrease (Supplementary Fig. 3). Nominal changes were observed after the development of mature biofilms (>72 h) and the communities approached a steady state. ASV1 *Pseudomonas* and ASV2 *Arthrobacter*

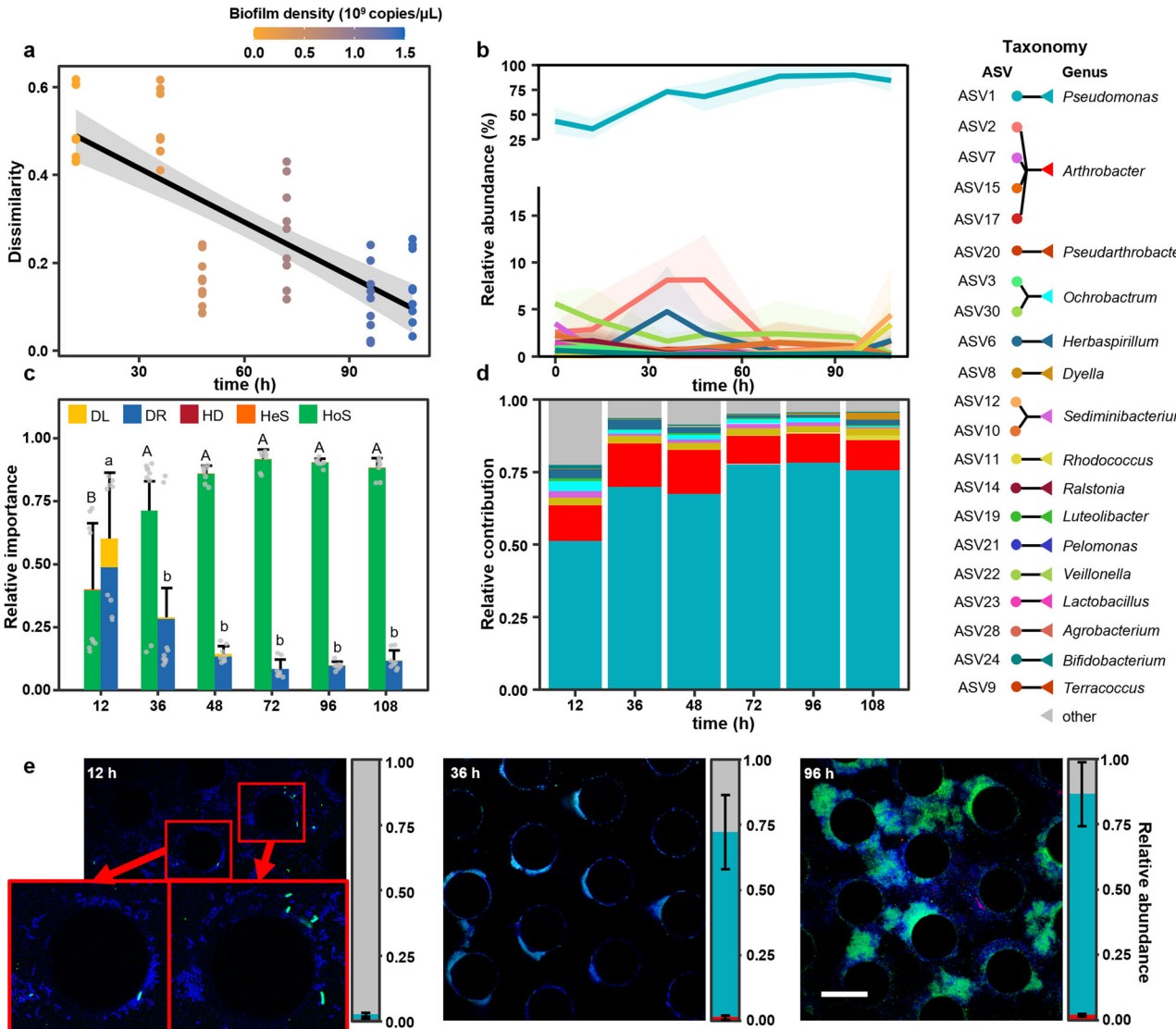

**Fig. 2 | The spatial niche partitioning during biofilm development. a** The Bray-Curtis dissimilarity of communities between adjacent time intervals is negatively correlated with the incubation periods (two-sided Pearson $r = -0.754$, $p = 4.69 \times 10^{-11}$). The solid line represents the linear regression, while the gray shading indicates the 95% confidence interval. **b** The relative abundances of the TOP 20 ASVs (covering 86.3% of total reads) in the total microbial community. Line colors correspond to different ASVs in the taxonomy legend at right. The solid line represents the average value and shaded areas indicate the standard deviation of three biological replicates. Source data are provided as a Source Data file. **c** The relative importance of different ecological processes in biofilm community assembly ($n = 9$ comparisons between three biologically replicates at each time point and three biological replicates for inoculum), including homogeneous selection (HoS), heterogeneous selection (HeS), homogeneous dispersal (HD), dispersal (DL) and drift (DR). Data are presented as mean values ± standard deviation of the relative importance of deterministic and stochastic processes. Different letters indicate significant differences ($p < 0.05$, one-way ANOVA). For exact $p$ values, see Supplementary Table 1. **d** The relative contribution of different genera to the community succession. Columns are colored based on the taxonomy (see the taxonomy legend on the right). **e** FISH images of biofilm in the microfluidic chamber. The biofilm is simultaneously hybridised with the probes for *Arthrobacter* (ART179-Alexa546, red) and *Pseudomonas* (PSE227-Alexa488, green). Biofilm cells are stained with DAPI (blue). The proportion of *Pseudomonas* and *Arthrobacter* in biofilm is calculated based on the area of green and red fluorescent cells ($n = 4$ chips × 15 grains). Data are presented as mean values ± standard deviation. Scale bar represents 50 μm.

were the two most abundant Amplicon Sequence Variants (ASVs) across the entire incubation period, accounting for 76.4% of total reads (Fig. 2b). The relative abundance of these two ASVs increased concurrently in the early stage (<48 h) but varied after mature biofilm formation (Fig. 2b).

The relative importance of different taxonomic groups to the total community succession was inferred via a phylogenetic-bin-based null model analysis (iCAMP)[18]. This framework divides observed ASVs into different taxonomic groups based on the phylogenetic relationships. The relative importance of ecological processes governing the turnovers of each group is determined by the null model analysis based on beta Net Relatedness Index ($\beta$NRI) and modified Raup-Crick metric (RC). The pairwise comparison with $\beta$NRI < −1.96 is an indicative of homogeneous selection, whereas that with $\beta$NRI > 1.96 interpreted as heterogenous selection. The taxonomic dissimilarity metric RC is applied for the pairwise comparisons with $|\beta$NRI$| \leq 1.96$. RC values >0.95 or <−0.95 represent homogenizing dispersal or dispersal limitation, while $|RC| \leq 0.95$ is interpreted as drift. The relative importance of individual processes is weighted by the relative abundance of each taxonomic groups and summed to estimate their relative importance in controlling community succession. The result reveals that the microbial community succession was driven by homogeneous selection, the relative importance of which increased from 39.5% to above 90.0% with biofilm development (Fig. 2c). At the genus level, *Pseudomonas* made the most prominent contribution to community succession, followed by *Arthrobacter* contributing more than 15% during the early stage (≤48 h) (Fig. 2d). These results demonstrate that after a temporary stochastic period (≤12 h), the microbial communities were driven by homogeneous abiotic and biotic environmental conditions and *Pseudomonas* and *Arthrobacter* were the key taxa shaping community structure.

## Spatial segregation during early biofilm development

To visually and spatially track the fate of *Pseudomonas* and *Arthrobacter* during biofilm development, FISH was performed in situ in the microfluidic chips with genus-specific probes. After the initial 12 h attachment, a minimal amount of surface-associated bacteria were identified as *Pseudomonas* and *Arthrobacter* and most of them still remained in the planktonic community (Fig. 2e and Supplementary Fig. 4). Remarkably, the proportion of *Pseudomonas* in the biofilm increased to 70.7 ± 14.2% at 36 h, during which its relative abundance in the planktonic community decreased simultaneously from 68.4 ± 5.2% to 18.5 ± 8.6% (Fig. 2e and Supplementary Fig. 4). Based on qPCR and amplicon sequencing analyses (Figs. 1d, 2b and Supplementary Fig. 4), 19.5 ± 12.7% of *Pseudomonas* in the microfluidic chamber inhabited biofilms at 12 h. The ratio further increased to 97.3 ± 10.0% at 36 h. The increased ratio of sessile *Pseudomonas* cells suggests that *Pseudomonas* underwent a transition from planktonic to biofilm-forming in the early stage of biofilm development. Conversely, *Arthrobacter* was still rarely observed in the biofilm after 36 h growth, while its proportion in the planktonic community increased from 15.8 ± 2.3% to 64.5 ± 10.4%, becoming the most prevalent taxon in the planktonic communities (Fig. 2e and Supplementary Fig. 4). The emergence of patches dominated by one genus represents the occurrence of spatial segregation in the early stage of biofilm development, through which free-living *Arthrobacter* and biofilm-forming *Pseudomonas* occupied the ambient fluid and grain surface, respectively. The dominance in biofilm increases the frequency of interactions between cells with similar genotypes[19–22], while the prevalent plankton taxa can exert a strong impact on the exometabolite pool[23,24]. The occurrence of spatial segregation is consistent with the elevated importance of selection in community assembly (Fig. 2c, e), indicating the selective force emerges through spatial organization.

In the late stage, densely packed *Pseudomonas* biofilms were observed to coat the grain and cover pore spaces, and the proportion

of *Pseudomonas* in the biofilm increased to 84.5 ± 12.3% at 96 h (Fig. 2e). The planktonic community was dominated by bacteria dispersed from the biofilm community, consisting mainly of *Pseudomonas*, *Rhodococcus* and *Lysinibacillus* (Supplementary Fig. 4). Given the simultaneous increase in relative abundance (Fig. 2b), we hypothesize that a potential positive interaction between *Pseudomonas* and *Arthrobacter* induces spatial segregation.

## Individual growth and pairwise interaction indicate *Arthrobacter* induces biofilm formation

To disentangle the underlying mechanisms in spatial segregation, we characterized the individual growth performance and pairwise interaction of bacterial strains isolated from the microfluidic chips (Fig. 3a). Twenty most abundant isolates from four different genera were chosen, which together accounted for 78.9% of the total abundance. Their planktonic growth and biofilm formation in microplates were assayed by optical density measurements and crystal violet staining, respectively. Although all the isolates showed substantial planktonic growth in the intensive soil extract medium (ISEM), *Arthrobacter* strains exhibited minimal biofilm-forming capability (Fig. 3a). This suggests that the extinction of *Arthrobacter* in the late stage of biofilm growth was attributed to its deficiency in biofilm formation and failure in permanent colonization in the presence of the continuous flushing with ISEM. Further, pairwise interaction experiments were carried out to test the hypothesis that a positive interaction between free-living *Arthrobacter* and biofilm inhabitants induced spatial segregation (Fig. 3a). The results reveal that most intergeneric interactions between *Arthrobacter* strains and the biofilm-forming isolates, *Pseudomonas* and *Rhodococcus*, were positive (Fig. 3a). 20 out of 25 co-culture combinations between *Arthrobacter* and *Pseudomonas* were assigned as positive interactions ($Y_{co} > Y_{sum}$).

Since the crystal violet assay quantified the total biofilm biomass in co-culture, the abundance of *Arthrobacter* and biofilm-forming isolates in plankton, biofilm and total community was quantified using qPCR to evaluate their individual fitness. The total cell numbers of *Arthrobacter* and biofilm-forming strains in 41 and 36 of 50 co-culture mixtures were significantly higher than those in monoculture, respectively (Supplementary Figs. 5 and 6). Consistent with the observed community spatial structure in microfluidic chambers (Fig. 2e), the co-culture biofilm was mainly composed of the biofilm-forming species (median frequency of biofilm-forming strains in biofilm = 0.605, Wilcoxon-signed rank test versus 0.5, $n = 50$ combinations, $p = 8.88 \times 10^{-5}$), and the plankton was dominated by *Arthrobacter* (median frequency of *Arthrobacter* strains in plankton = 0.878, Wilcoxon-signed rank test versus 0.5, $n = 50$ combinations, $p = 7.79 \times 10^{-10}$). After 24 h growth, the frequencies of *Arthrobacter* in all the 50 co-culture combinations were comparable to the initial proportions (Wilcoxon-signed rank test versus *Arthrobacter* frequency in inoculum, $n = 4$ replicates for each combination, $p > 0.05$), which suggested that *Arthrobacter* and biofilm-forming species didn't exclude each other in co-culture.

Integrated genomic and transcriptomic analyses of two co-culture combinations (ASV2 *A. ramosus*-ASV1 *P. fluorescens* and ASV2 *A. ramosus*-ASV11 *R. erythropolis*) were performed to reveal the transcriptional responses in interspecific interactions. Co-culture with *A. ramosus* induced the differential expression of 2165 and 161 genes in *P. fluorescens* and *R. erythropolis*, respectively. Based on Gene Set Enrichment Analysis (GSEA), the metabolic pathways including oxidative phosphorylation, carbon metabolism, ribosome and citrate cycle of *P. fluorescens* were activated in co-culture (Supplementary Fig. 7). Besides upregulated metabolic pathways, biofilm formation-related genes of *P. fluorescens* were highly expressed. Both the biosynthesis genes for intra-and intercellular biofilm signaling molecules, i.e., c-di-GMP and quinolone signal (PQS), were significantly upregulated in co-culture (Fig. 3b). The activation of biofilm signal synthesis

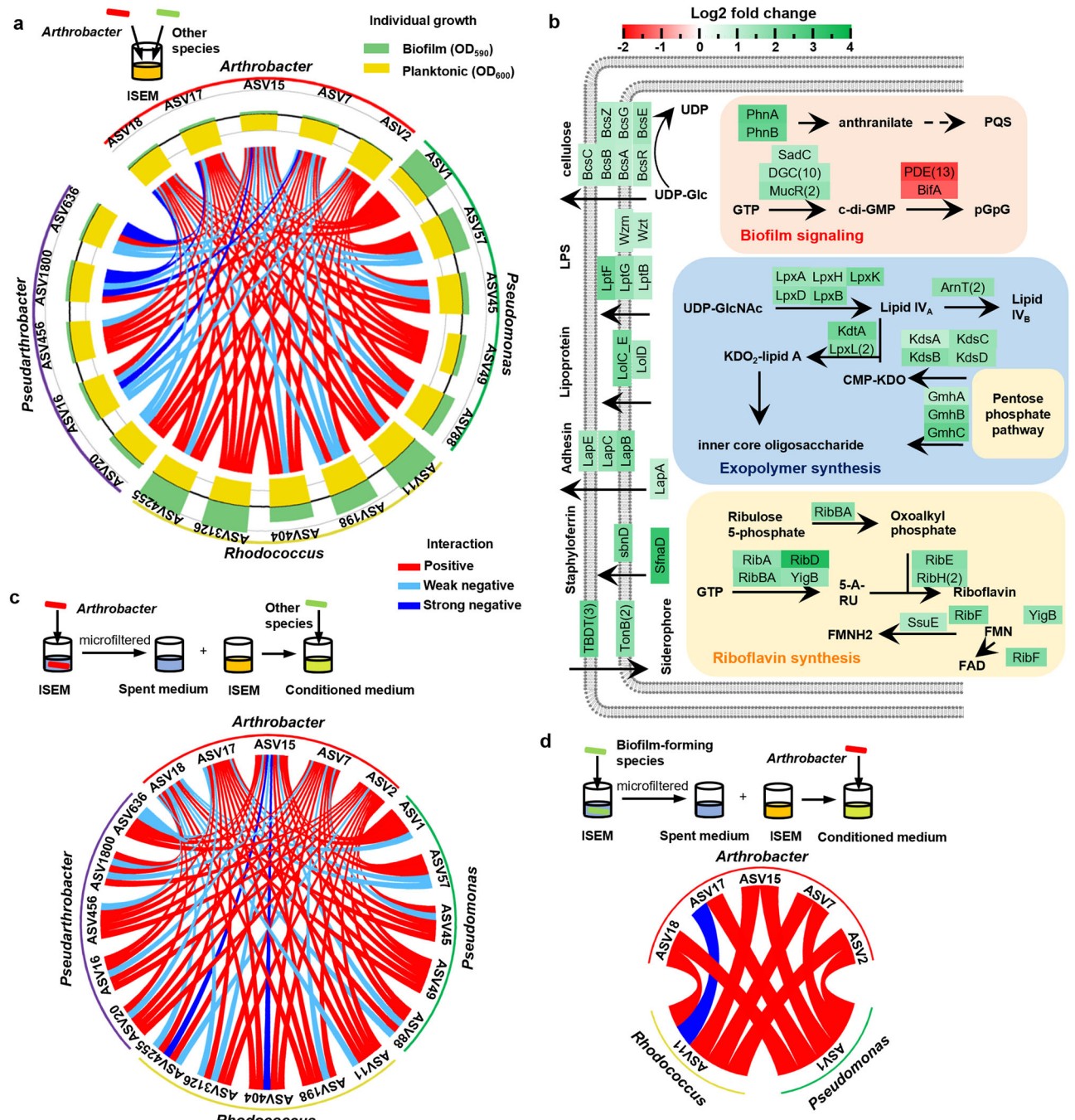

**Fig. 3 | The individual growth performance and pairwise interaction in ISEM.**
**a** The individual growth and pairwise co-culture interaction between *Arthrobacter* strains and isolates from other 3 genera in ISEM. Individual planktonic growth is measured by $OD_{600}$. The biofilm yield is quantified by $OD_{590}$ after crystal violet staining. Based on the minimum ($Y_{min}$), average ($Y_{ave}$), maximum ($Y_{max}$) and sum ($Y_{sum}$) monoculture biofilm yield of each member in co-culture and the co-culture biofilm yield ($Y_{co}$), the pairwise interaction can be classified as positive ($Y_{co} > Y_{sum}$), strong negative ($Y_{co} < Y_{min}$) or weak negative ($Y_{sum} \geq Y_{co} \geq Y_{min}$). **b** Key differentially expressed biofilm-formation related genes of ASV1 *P. fluorescens* in co-culture with

ASV2 *A. ramosus*. Numbers in brackets represent the number of differentially expressed genes with the same functions. Source data are provided as a Source Data file. The interaction in ISEM conditioned by planktonic *Arthrobacter* (**c**) and biofilm-forming isolates (**d**). Based on the biofilm or planktonic growth in conditioned medium ($Y_c$) and unconditioned medium ($Y_u$), the interaction can be classified as positive ($Y_c \geq Y_u$), weak negative ($1 > Y_c/Y_u \geq 0.5$) or strong negative ($Y_c/Y_u < 0.5$). Source data for Fig. 3a, c are provided as a Source Data file. The measured $OD_{600}$ for Fig. 3d was plotted in Supplementary Fig. 10a.

corresponded to the upregulated synthesis and export pathways for exopolymers including adhesin LapA, extracellular polysaccharide and lipopolysaccharide. Moreover, the genes of *P. fluorescens* responsible for the synthesis of cooperative public goods like siderophore (staphyloferrin) and electron shuttles (riboflavin (RF), flavin mononucleotide (FMN) and flavin adenine dinucleotide (FAD)) were significantly upregulated to provide a collective benefit (Fig. 3b). GESA

analysis revealed that co-culture also upregulated amino acid metabolism and secondary metabolite biosynthesis in *R. erythropolis* (Supplementary Fig. 7). Each of these two biofilm-forming strains activated 4 different amino acid metabolic pathways. A total of 237 and 263 genes involved in the biosynthesis of cofactors and secondary metabolites were positively enriched in *P. fluorescens* and *R. erythropolis*, respectively. The induced biofilm formation was validated in

the microfluidic chamber. *A. ramosus* induced a 2.3-and 4.4-fold increase in the biofilm thickness of *P. fluorescens* and *R. erythropolis*, respectively (Supplementary Fig. 8). Meanwhile, only 126 and 88 genes of strain *A. ramosus* were differentially expressed in co-culture with *P. fluorescens* and *R. erythropolis* compared to the monoculture (Supplementary Fig. 9). In these two co-culture systems, *A. ramosus* shared upregulated iron acquisition genes like *efeUOB* and downregulated genes for peptide/nickel transport.

As free-living *Arthrobacter* and biofilm-forming bacteria were spatially segregated in the porous environment (Fig. 2e and Supplementary Fig. 4), we suppose that the cooperative interaction was mediated via extracellular metabolites. To test this, biofilm-forming isolates were grown in a conditioned medium, prepared by mixing fresh ISEM and *Arthrobacter* spent medium in a 1:1 ratio[25] (Fig. 3c). *Arthrobacter* spent medium was the cell-free culture supernatant of *Arthrobacter* grown in ISEM for 48 h. In agreement with the co-culture experiments, the medium conditioned by *Arthrobacter* strains substantially enhanced biofilm formation of *Pseudomonas* and *Rhodococcus* (Fig. 3c). On the other hand, ISEM conditioned by ASV1 *P. fluorescens* and ASV11 *R. erythropolis* significantly enhanced the growth of most *Arthrobacter* strains (Fig. 3d and Supplementary Fig. 10a). The enhanced growth was also observed in M9 minimal medium conditioned by ASV1 *P. fluorescens* and ASV11 *R. erythropolis* (Supplementary Fig. 10b), which suggested *Arthrobacter* strains benefited from the metabolites secreted by the biofilm-forming strains. As public goods such as siderophore, FMN and FAD were detected in the supernatant of ASV1 *P. fluorescens* grown in ISEM (Supplementary Figs. 11 and 12), we further investigated whether *Arthrobacter* exploited these public goods by constructing the siderophore synthesis mutant in ASV1 *P. fluorescens* (ΔsfnaD) and adding flavins to the culture media. Deletion of the siderophore synthetase gene reduced the growth of most *Arthrobacter* strains in the spent medium of ASV1 *P. fluorescens* (Supplementary Fig. 13). Meanwhile, the supplementation of flavins at the same level as the ISEM conditioned by ASV1 *P. fluorescens* significantly enhanced the growth of *Arthrobacter* in both ISEM and M9 medium (Supplementary Fig. 14). These results support our hypothesis that excreted metabolites of free-living *Arthrobacter* can induce spatial segregation by promoting biofilm formation, which simultaneously enhances the fitness of *Arthrobacter* in planktonic communities via public goods production.

### D-amino acid (DAA) consumption triggers spatial segregation

Towards understanding exometabolite-driven spatial segregation, the dynamics of extracellular metabolites in the effluents of microfluidic chips were unraveled via untargeted metabolomics analysis. A total of 162 metabolites were identified, among which amino acids (AAs) and their derivatives (24.0%) and fatty acids (13.0%) were the main constituents. Procrustes analysis revealed a strong and highly significant correlation between exometabolites and the total community composition in the microfluidic chips (Fig. 4a). In line with community structures, exometabolite profiles in the early stage were distinct from that of late stage. All the measured exometabolites were divided into three clusters based on their dynamics during biofilm development, including released (cluster 1), consumed (cluster 2) and the others (cluster 3) (Supplementary Figs. 15–18). Putative carbohydrates (for example lactose) and purines were rapidly consumed after inoculation (Supplementary Fig. 17). A number of putative AAs and their derivatives, grouped in cluster 1, accumulated during biofilm development (Supplementary Fig. 16). Spearman's rank correlation was computed to evaluate the directionality of microbe-AA relationships (Fig. 4b). During the entire incubation period, 11 of 18 identified amino acids were negatively correlated with the relative abundance of *Arthrobacter* and 8 of them displayed a positive relationship with *Pseudomonas*. This implies that these two dominant genera play different roles in the amino acid metabolism.

As metabolomics may yield false positives in metabolite identification, the concentrations of free amino acids and their enantiomers were determined via spectroscopic detection after derivatization. The initial total concentration of free amino acids in ISEM was 63.7 mg/L (Fig. 4c), mainly consisting of Thr, Val, Met, Leu, Phe, His and Lys. Consistent with the untargeted metabolomic analysis, the total amino acid concentration increased substantially in the late stage, to a final concentration above 300 mg/L. Meanwhile, the dynamics of DAAs exhibited a "V-shaped" pattern which decreased upon inoculation and accumulated after 48 h growth (Fig. 4d). The consumption of D-amino acid aligned well with the growth of *Arthrobacter*, suggesting a potential role of *Arthrobacter* in D-amino acid consumption. Since D-amino acids are known to suppress biofilm formation via inhibiting initial attachment, EPS production and quorum sensing[26–29], we propose that D-amino acid consumption is the central metabolic trait to trigger spatial segregation.

### *Arthrobacter* induces biofilm formation via DAA hydrolysis

The isolated strains were cultivated in ISEM to probe the DAA removal capability. The initial DAA concentration of ISEM was 44.1 mg/L. *Arthrobacter* strains removed about 90% of DAA within 48 h and displayed the highest DAA consumption capacity (Fig. 5a). Meanwhile, ~60% DAA concentration remained in the *Pseudomonas* culture. Compared with that of ISEM, DAA concentration in the ISEM conditioned by *Arthrobacter* decreased to ~25 mg/L (fresh ISEM mixed with the supernatant of *Arthrobacter* culture at a ratio of 1:1). With reduced DAA concentration, the biofilm formation of *Pseudomonas* and *Rhodococcus* in the conditioned medium was enhanced compared to that in ISEM (Fig. 3c). To validate the role of DAAs in biofilm formation, various amounts of DAAs were supplemented into the conditioned medium. Biofilm formation was found to be inhibited with the increase of DAA concentration. A concentration of 45 mg/L DAAs in conditioned media, which was equivalent to the level of ISEM, was found to cause 31.8 ± 7.9% and 27.7 ± 6.4% reduction in *Pseudomonas* and *Rhodococcus* biofilm formation (Fig. 5b). Overall, these results strongly support our hypothesis that *Arthrobacter* induces spatial segregation via removal of the biofilm inhibitor, DAAs.

## Discussion

Subsurface porous environments such as soil and sediment are one of the major microbial habitats on Earth[1,2]. In these environments, biofilm formation is a fundamental living strategy for microorganisms. The intricate interactions within local communities give rise to community diversity and stability. The role of microbial interactions in shaping complex natural biofilm community successions, however, is rarely examined. Here, using a microfluidic chip environment, we demonstrate that a positive microbial interaction drives the community spatial structure during biofilm colonization. Biofilm-deficient or more planktonic *Arthrobacter* triggers biofilm formation via removal of biofilm inhibitors. With the decreased level of biofilm inhibitors, *Pseudomonas*, the most abundant genus in the total community, reduced its proportion in the ambient fluid and switched to a sessile lifestyle. The spatial segregation was also observed in co-culture containing *Arthrobacter* and *Pseudomonas* strains with comparable initial cell densities. The pairwise interaction analyses revealed that *Arthrobacter* and biofilm-forming strains dominated the plankton and biofilm in the co-culture systems, respectively. The proportion of *Pseudomonas* that inhabited biofilms in co-culture was significantly higher than that in monoculture ($p < 0.001$, one-way ANOVA, Supplementary Table 6). Although the growth in ISEM and M9 minimal medium with different DAA levels suggests that *Arthrobacter* doesn't gain fitness benefits through DAA hydrolysis (Supplementary Fig. 19), it receives return benefits from biofilm-forming species which secrete public goods. The mutually beneficial interaction facilitates the occupation of segregated niches by free-living *Arthrobacter* and biofilm-forming

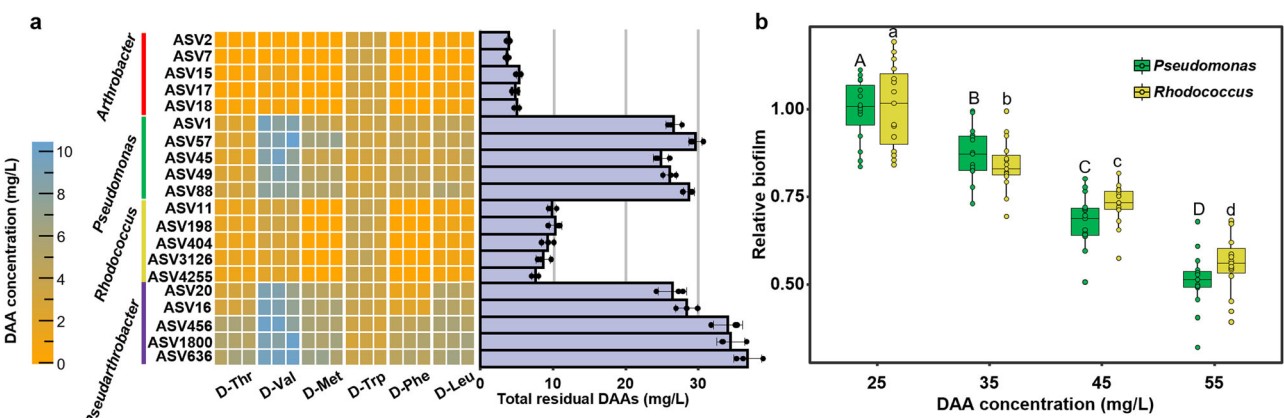

**Fig. 4 | The dynamics of the exometabolite pool during community assembly.**
**a** Procrustes rotation of the total microbiota composition nonmetric multi-dimensional scaling (NMDS) plot with the metabolome NMDS plot (Procrustes correlation = 0.72, $p$ = 0.001). **b** Correlation between the ASVs (relative abundance >0.1% across total microbial communities in microfluidic chips) and amino acids. Only those with Spearman's rank correlation coefficients ($|r| \geq 0.5$, ***$p \leq 0.001$,

\*\*$p \leq 0.01$, \*$p \leq 0.05$) are shown. ACC, 1-Aminocyclopropane 1-carboxylic acid; Ect, Ectoine; 2-Akbt, 2-amino-3-oxobutanoic acid. **c** The accumulation of total amino acids in the supernatant ($n = 3$ chips). **d** The dynamics of DAA during biofilm development exhibit a V pattern ($n = 3$ chips). The shaded areas represent the standard deviation of three biological replicates. Source data are provided as a Source Data file.

**Fig. 5 | The inhibitory effect of DAAs on biofilm formation. a** The residual DAAs in ISEM after 48 h growth of each isolate ($n = 3$ biologically independent replicates). The bar chart shows the total residual DAAs in the culture. Data are presented as mean values ± standard deviation. **b** The relative biofilm formation in the conditioned medium with different levels of DAAs ($n = 5$ isolates for each genus × 3

replicates). The biofilm biomass of each isolate, indicated by OD$_{590}$, is normalized to that developed in the conditioned medium with 25 mg/L D-amino acid. Different letters indicate significant differences ($p < 0.05$, one-way ANOVA). For exact $p$ values, see Supplementary Table 5. The boxplots display the 1.5 × interquartile range (whiskers), quartiles, and median of relative biofilm biomass.

bacteria and shapes community spatial organization. It indicates that the planktonic microbial minorities can impose strong selection pressure on biofilm community succession, by serving as inhibitor scavengers and, in turn, enhance their own fitness in the pore fluid.

Simulated and experimental results have demonstrated that spatial segregation is generally an outcome of competitive interactions in structured environments[5,30,31]. As competing microorganisms reciprocally exclude one another, spatial segregation contributes to the stable coexistence of competitors and increases community diversity[12,32]. In particular, biofilm-forming species gain competitive advantages over free-living or biofilm-deficient species in natural environments. Previous studies found biofilm-forming species outcompete free-living cells via smothering them or cutting off nutrient access[33,34]. By contrast, we show that free-living species, which are hitherto recognized as the outcompeted species, can in fact establish mutualistic cooperation with biofilm-forming cells. In the early stage of colonization, the free-living microbes scavenge the universal biofilm inhibitor to induce biofilm formation. With reduced inhibitory effect, biofilm-forming cells depart from ambient fluid to colonize grain surfaces and secrete public goods in return. These early surface settlers can rapidly preempt or modify the microhabitats, which impacts the colonization of late-arriving species and drives subsequent community succession[8,35,36]. This indicates that the cooperative interaction between free-living and biofilm-forming partners may hold a central role during the colonization of virgin territories.

We find that DAAs are the key metabolites that mediate spatial segregation in our system. The changes in DAAs during biofilm colonization reveal a V-shaped pattern. At the late stage, the frequency of *Arthrobacter* capable of efficiently hydrolyzing DAAs decreased significantly due to its deficiency in biofilm formation. The mean travel time of colloids in the microfluidic chamber decreased from $51.48 \pm 2.16$ to $30.72 \pm 5.16$ min after the biofilm development (two-tailed Student's *t*-test, $n = 3$ independent replicates for each condition, $p = 0.011$, Supplementary Fig. 2), indicating an accelerated elution of planktonic *Arthrobacter* at the late stage. The extinction of *Arthrobacter* in the porous environments led to the diminished metabolic capabilities of the remaining microbial community towards DAAs, the concentration of which in the effluent gradually increased to the original level in ISEM (Fig. 4d). As a common group of secondary metabolites and major constituents of bacterial walls, these amino acid enantiomers can be actively excreted and passively released to surrounding environments[37,38]. Micromoles of DAA per gram dry weight were detected in sediment and soil[37,39,40]. As DAAs over this concentration range are sufficient to suppress biofilm formation[26,27], they may therefore play a pervasive role in governing biofilm community structure over a wide range of ecosystems.

This study provides a mechanistic understanding of how microbial interactions may govern biofilm community succession in subsurface porous environments. Further efforts are required to address the roles of different biotic interactions, such as competition and mutualism in shaping the ecological stability and functions of complex biofilm communities, and to validate these findings in real soils and microbial communities. This experimental platform can be adapted to enable in situ visualization and integrate multiple omics technologies to elucidate spatiotemporal dynamics of microbial activities and underlying mechanisms, which will advance our understanding of the spatial organization of microbial communities and functional traits in ecosystems.

## Methods

### Extracting microorganisms from soil

Soil was sampled from a paddy field of the National Agro-Ecosystem Observation and Research Station (Jiangxi Province, China, 116°55′E, 28°15′N) to a depth of 0–20 cm, in Nov 2017. Soil samples were ground

to pass through a 10-mesh sieve and microorganisms were extracted using Nycodenz gradient centrifugation[41].

### Growth media

Intensive soil extract medium (ISEM) based on soil extract was prepared using the same soil sample for soil microorganism extraction following previous work[42]. Briefly, 500 g dry soil was mixed with 1.3 L 80% methanol and stirred overnight at room temperature. The supernatant was collected and residual soil was extracted a second time. The two supernatants were combined, passed through a cellulose filter paper and subject to lyophilization. The lyophilized sample was solubilized in 200 mL water and filtered through a 0.22 μm filter. The filtrate was designated as soil extract solution. Each liter of ISEM consisted of 0.2 L soil extract solution, 0.23 g $KH_2PO_4$, 0.23 g $K_2HPO_4$, 0.23 g $MgSO_4 \cdot 7H_2O$, 0.33 g $NH_4NO_3$, 0.25 g $NaHCO_3$, 5 mg of each D-amino acids (D-valine, D-methionine, D-leucine, D-phenylalanine, D-threonine, and D-tryptophan), 1 mL of the vitamin stock solution (thiamine hydrochloride, 0.5 g/L; riboflavin, 0.5 g/L; niacin, 0.5 g/L; pyridoxine HCl, 0.5 g/L; inositol, 0.5 g/L; calcium pantothenate, 0.5 g/L; β-aminobenzoic acid, 0.5 g/L; biotin, 0.25 g/L), 2 mL of the selenite-tungstate solution (NaOH, 0.5 g/L; $Na_2SeO_3 \cdot 5H_2O$, 3 mg/L; $Na_2WO_4 \cdot 2H_2O$, 4 mg/L) and 2 mL of the trace element solution SL-10 (HCl, 2.8 g/L; $FeCl_2 \cdot 4H_2O$, 1.5 g/L; $ZnCl_2$, 70 mg/L; $MnCl_2 \cdot 4H_2O$, 100 mg/L; $H_3BO_3$, 6 mg/L; $CoCl_2 \cdot 6H_2O$, 190 mg/L; $CuCl_2 \cdot 2H_2O$, 2 mg/L; $NiCl_2 \cdot 6H_2O$, 24 mg/L; $Na_2MoO_4 \cdot 2H_2O$, 36 mg/L)[43,44]. The pH of ISEM was adjusted to 6.8 by addition of HCl or NaOH. Three batches of ISEM were prepared to assess the batch-to-batch variations. On the basis of three-dimensional excitation–emission matrix (3D-EEM) and UV-vis spectroscopic analyses, the composition of ISEM was consistent across the independent batches (Supplementary Fig. 20). M9 minimal medium was prepared to evaluate the influence of DAA on microbial growth. The minimal medium consisted of 2.0 g/L glucose, 6.0 g/L $Na_2HPO_4$, 3.0 g/L $KH_2PO_4$, 0.5 g/L NaCl, 1.0 g/L $NH_4Cl$, 0.12 g/L $MgSO_4$, 11.1 mg/L $CaCl_2$, 3.49 μg/L $(NH_4)_6Mo_7O_{24}$, 24.73 μg/L $H_3BO_3$, 3.90 μg/L $CoCl_2$, 1.60 μg/L $CuSO_4$, 10.07 μg/L $MnCl_2$, 1.61 μg/L $ZnSO_4$ and 0.15 mg/L $FeSO_4$.

### Microfluidic chip fabrication

The microfluidic chamber contains a matrix of pillars, each 50 μm in diameter and height. The microchannel design was exposed onto a silicon wafer via a laser writing system (Microwriter ML3). Deep Reactive Ion Etching was carried out in the AZ400K:H20 (1:4) solution to generate the microfluidic structure. The etched silicon wafer served as a master mold for casting polydimethylsiloxane (PDMS) channels. The patterned PDMS was bonded to a glass slide after plasma treatment. The microfluidic chips were sterilized with 75% ethanol prior to inoculation.

### Biofilm development in microfluidic chips

The microfluidic chip comprised a flow channel with one inlet and one outlet. The inlet tubing connected the microfluidic chip to a syringe which was mounted on a syringe pump to feed fresh ISEM for microbial growth. The effluent containing free living (planktonic) cells and exometabolites was collected at the outlet. The continuous flow system provided a well-defined and constant condition for biofilm development and enabled the continuous monitoring of planktonic community and exometabolite dynamics. Microbial cells isolated from soil were resuspended in phosphate buffered saline (PBS) to a final $OD_{600}$ of 0.1. 200 μL of soil bacterial suspension was introduced into microfluidic chips through an inoculation port located immediately downstream of the medium inlet (Supplementary Fig. 21a). The inlet tubing was clamped prior to inoculation to prevent backflow. The stainless steel needle was removed after injecting the inoculum followed by immediate sealing of the inoculation port with silicon glue (Supplementary Fig. 21b). After a one-hour attachment, ISEM was

supplied with a constant flow rate of 0.5 μL/min. The microfluidic chips were incubated at 25 °C in the dark. The stainless steel needle and tubing at the outlet were replaced with sterile ones before sampling. The effluent of microfluidic chips was collected to determine the cell density of planktonic communities. Thereafter, two sterile syringes were connected to the inlet and outlet of the microfluidic device. 1 mL of PBS buffer was repeatedly flushed back and forth between these two syringes at a flow rate of about 1,200 μL/min for twenty times to extract all bacterial cells in the microfluidic chamber. The amount of planktonic and the total microbial cells in the microfluidic chips were quantified by qPCR using the primer pair Eub338F (ACT CCT ACG GGA GGC AGC AG) and Eub518R (ATT ACC GCG GCT GCT GG). To construct the standard curve for qPCR, the PCR product from the model organism *P. putida* KT2440 was cloned into pMD8-T vector and electroporated into *E. coli* DH5α competent cells. Serial dilutions of the plasmid DNA were then amplified using the SYBR Green PCR Supermix (BioRad). The amplification efficiency determined based on the standard curve was 104.7%. The specificity of the amplified product was confirmed by melt curve analysis and gel electrophoresis (Supplementary Figs. 22 and 23). The amount of biofilm cells was calculated as the difference between the total and planktonic cell numbers:

$$C_{planktonic} = D_{planktonic} \times V_{chip} \qquad (1)$$

$$C_{biofilm} = C_{entire} - C_{planktonic} \qquad (2)$$

where $C_{planktonic}$ and $C_{biofilm}$ are the planktonic and biofilm cell numbers in the microfluidic chamber, $D_{planktonic}$ is the cell density in the effluent of the microfluidic chamber, $V_{chip}$ is the volume of the microfluidic chamber (3.0 μL), and $C_{entire}$ is the total cell number in the PBS eluate.

## Colloid transport and retention in the microfluidic chamber

To evaluate the residence time of planktonic cells in microfluidic chamber, we characterized the transport and retention of colloidal microspheres in the porous medium. The red fluorescent polystyrene microspheres purchased from Jiangsu Zhichuan Technology Co. (China) have a diameter of 2 μm and a density of 1.05 g/cm³. 3 μL of PBS buffer containing the fluorescent microspheres ($1.0 \times 10^5$ particles/μL) were pumped into the microfluidic chamber at a flow rate of 0.5 μL/min. 1.5 μL of effluent was collected every 3 min at the outlet using a pipette and then diluted in 50 μL of PBS buffer in 384-well microplates. The particle density in the effluent was determined by measuring fluorescent intensity (excitation, 370 nm; emission, 610 nm) (Supplementary Fig. 2c). Three independent replicates were carried out. The mean travel time ($\tau$) was calculated as follows[45,46]:

$$\tau = \frac{\int_0^T t C(t) dt}{\int_0^T C(t) dt} \qquad (3)$$

where $T$ is the duration of experiments, $C(t)$ is the particle density in the effluent at time point $t$.

## Confocal microscopy and image processing

Biofilms in microfluidic chambers were stained using fluorescent dyes and observed under a confocal microscope (A1R, Nikon). Microbial cells were stained with DAPI or SYTO 9 for 30 min. Both fluorescent dyes bind to genomic DNA and produce overlapping fluorescence (Supplementary Fig. 24). Confocal images were analyzed by a custom-written MATLAB code. After image binarization, the biofilm thickness on each grain was computed as the average radial distance from points on the biofilm edges to the grain surface. Biofilm roughness was represented by the standard deviation of biofilm thickness on grain surfaces[46].

## Fluorescent in situ hybridization in microfluidic chip

PSE227-Alexa488 (5′-AAT CCG ACC TAG GCT CAT C-3′) was used to visualize the distribution of *Pseudomonas* in the biofilm[47]. To design a FISH probe for *Arthrobacter* strains, a consensus sequence for 292 *Arthrobacter* isolates from the microfluidic chamber was generated using Usearch and the probe sequence was designed via primer 3[48,49]. Alexa546-labled ART179 (5′-CAT GCG TGG AGC GGT CGT-3′) was used to trace *Arthrobacter*. The probes were validated with pure cultures (Supplementary Figs. 25 and 26). The universal bacterial probe EUB338 (5′-GCT GCC TCC CGT AGG AGT-3′) and non-sense probe (5′-ACT CCT ACG GGA GGC AGC-3′) were used as the positive and negative control, respectively. All the FISH probes were synthesized by Thermo Fisher Scientific (Guangzhou, China). The specificity and sensitivity of FISH were assessed by calculating the detection rate, which is the proportion of DAPI-stained cells detected by FISH. The high detection rates of positive samples and few nonspecific false positive signals indicate that the FISH analysis is a feasible and reliable approach for quantifying *Pseudomonas* and *Arthrobacter* (Supplementary Fig. 27). FISH in microfluidic chambers was performed[50]. All reagents were delivered into the microfluidic chamber at a flow rate of 0.5 μL/min. The biofilm was initially fixed in 2% formaldehyde solution for 1.5 h and washed with PBS buffer for 40 min. The formaldehyde fixation preserves biofilm integrity for subsequent sample pretreatment (Supplementary Fig. 28). The fixed biofilm sample was then permeabilized with 10 mg/mL lysozyme for 40 min at 37 °C, followed by a flush step. Dehydration was performed by flowing 50%, 80% and 98% ethanol solution through the chamber for 20 min, respectively. 2 mL of the hybridization buffer included 600 μL of formamide, 998 μL of water, 360 μL of 5 M NaCl, 40 μL of 1 M Tris-HCl and 2 μL of 10% SDS. The FISH probe was added to the hybridization buffer to a final concentration of 2.5 ng/μL. The hybridization buffer was loaded into the chamber over 30 min and left for 3 h at 48 °C. Then the sample was washed with the washing buffer (20 mM Tris-HCl, 102 mM NaCl, 5 mM EDTA and 0.01% SDS) at 48 °C for 40 min prior to microscopic observation.

## 16S rRNA gene amplicon sequencing

To investigate the microbial community composition in microfluidic chambers, the microbial cells in microfluidic chambers were flushed out with PBS buffer. Total DNA was extracted using the EZNA soil DNA kit (Omega). The universal primers 338F/806R were used for 16S rRNA gene amplification[51]. The amplification was carried out using Q5 High-fidelity DNA polymerase (NEB). The PCR program comprised 2 min initial DNA denaturation at 98 °C; 25 cycles at 98 °C for 15 s, 55 °C for 30 s and 72 °C for 30 s; and 30 s extension at 72 °C. The amplicons were purified using VAHTS DNA Clean Beads (Vazyme) and then sequenced on the Illumina Miseq sequencing platform with 300 bp paired-end reads. At each sampling point, the microbial communities including biofilm and planktonic cells were collected from three independent microfluidic chips. Planktonic communities were assessed by collecting the effluent of microfluidic chips.

## Exometabolomic analyses

To elucidate the dynamics of extracellular metabolites, the effluent of the microfluidic chips was subject to metabolomic analyses. 100 μL of effluent was collected from each of six independent microfluidic chips at each time point. The effluent was filtered through a 0.2 μm microcentrifuge PVDF filter to remove bacterial cells. The filtrate was vortexed with 400 μL pre-cooled methanol/acetonitrile (50/50 v/v), left at −20 °C for 30 min and then centrifuged at 14,000 × g for 20 min. The supernatant was lyophilized and resuspended in 100 μL of 50:50 acetonitrile/water solution. After centrifuged at 14,000 × g for 15 min, the supernatant was harvested for LC-MS/MS measurement. Quality control (QC) samples were prepared by pooling equal aliquots from all 42 effluent samples collected during the incubation period. Five aliquots of QC

samples were injected prior to the analysis and after every ten runs to access the analytical variance (Supplementary Figs. 29 and 30). Untargeted metabolomics was performed with Ultimate 3000 UHPLC system fitted with a Q-Exactive Orbitrap mass spectrometer. The mobile phase A is 0.1% formic acid and mobile phase B is acetonitrile. The solvent gradient conditions were as follows: 0–1.0 min, 95% A; 1.0–9.0 min, 95 ~ 0% A; 9.0–12.0 min, 0% A; 12.0–15.0 min, 95% A, with a flow rate of 0.3 mL/min. Chromatography was performed on an ACQUITY UPLC BEH Amide column (Waters, 1.7 μm, 2.1 mm × 100 mm). The ESI source temperature was 320 °C and the spray voltage was set at 3.5 kV. Mass spectra scans were collected from $m/z$ 80 to 1200 with a 200-ms accumulation time. MS/MS acquisition were preformed using information-dependent acquisition (IDA) mode. The collision energy was set at $35 \pm 15$ eV.

### Metabolite identification
The raw MS data were converted to mzXML format using MSConvert (version 3.0) and were subsequently processed by XCMS (version 3.2) for peak picking and alignment[52]. Metabolite identification was performed by accurate mass and MS/MS matching against METLIN, MassBank, LipidMaps, and mzCloud reference libraries[53–55]. The MS/MS spectral similarity was represented using the cosine score[56]. The XCMS settings were as follows: method = "centWave", ppm = 15, peakwidth = c(10,60), mzwid = 0.025, minfrac = 0.5, and bw = 5. Only metabolites at level 2 identification (putatively annotated compounds) were used for downstream statistical analysis. Based on KEGG annotation, the metabolites involved in amino acid metabolism were classified as amino acid intermediates.

### Bioinformatic analysis
The raw sequencing data was processed using the DADA2 (version 2021.2.0) in QIIME2 (version 2021.2) to merge and denoise paired-end reads[57,58]. Taxonomy was assigned using Naive Bayes classifier (version 2021.2.0) and the Greengenes database (version 13.8)[59]. The phylogenetic tree was built using the align-to-tree-mafft-fasttree pipeline[60,61]. The peak areas of exometabolite ions were transformed into z scores. All the exometabolites were grouped into three clusters based on Spearman's rank correlation between the incubation periods and the z score of each metabolite (Supplementary Fig. 15). Exometabolites with a correlation coefficient >0.5 were considered as "released" and those with a coefficient less than −0.5 were considered as "consumed"[62]. The remaining metabolites were categorized as "others".

### Null model analysis
Microbial community assembly was evaluated via null model analysis. The community assembly mechanisms were quantitatively inferred by a phylogenetic-bin-based null model analysis (iCAMP) using package "iCAMP" (version 1.3.4)[18]. The analysis was conducted using the parameters recommended by the developers[18]. The threshold of phylogenetic distance was set to 0.2 and the minimal bin size was 12. The 11,985 observed ASVs were classified into 442 different phylogenetic bins. The null model distribution was generated using 1000 randomizations[63]. The null model algorithm "taxa shuffle" was applied to shuffle the taxa across the phylogenetic tree and randomize the phylogenetic relationship within phylogenetic bins[64]. The $\beta$-Net Relatedness Index ($\beta$NRI) and taxonomic $\beta$-diversities using modified Raup−Crick metric (RC) were determined to identify the ecological process governing each bin. The cutoffs of significant $\beta$NRI value, significant RC value and significant one-side confidence level were 1.96, 0.95 and 0.975, respectively. The relative importance of different ecological processes in the turnover between communities in microfluidic chips and inoculums and the contributions of individual phylogenetic bin to ecological processes were calculated.

### AA quantification
The bacterial culture and the effluent were filtered through 0.2 μm filters prior to analysis. The total free AA concentration was determined via A300 amino acid analyzer (membraPure). The system was calibrated with a certified standard amino acid mixture of 17 proteinogenic amino acids (GBW(E)100062). The standard mixture was injected at the beginning of the analysis and after every 10 samples to evaluate the stability of the analytical platform. The free AA was quantified using HPLC with fluorescence detection after pre-column derivatization with o-phtalaldehyde (OPA) and N-isobutyryl-L-cysteine (IBLC)[65]. 0.5 μL of sample was mixed with 2.5 μL of 0.4 M borate buffer and then reacted with 0.25 μL of derivation reagent (260 mM IBLC and 170 mM OPA in 0.4 M borate buffer). After dilution with 15 μL of 0.1% acetic acid, 15 μL of the mixture was subject to HPLC analysis. The separation of AA derivatives was performed on an Agilent Poroshell HPH-C18 column at 30 °C with a flow rate of 0.7 mL/min. A dual gradient elution was conducted with the mobile phase consisting of 50 mM sodium acetate (A) and acetonitrile/methanol/water (45/45/10) (B). The solvent gradient was as follows: 0–2 min, 4% B; 2–4 min, 10% B; 4–15 min, 20% B; 15–27 min, 35% B; 27–35 min, 50% B; 35–37, 100% B and held for a further 5 min at 100% B. The elute was monitored using a fluorescence detector (excitation at 230 nm and emission at 450 nm). Calibration curves were constructed for each AA and their enantiomers (Supplementary Fig. 31). QC samples were prepared at 1.0, 4.0 and 9.0 mg/L to assess the method precision and accuracy, which expressed as the relative standard deviation and bias, respectively (Supplementary Table 8). Three replicates of each QC level were analyzed. The relative standard deviation (RSD, %) was calculated as standard deviation/mean × 100. The bias (%) was determined as (measured concentration−theoretical concentration)/theoretical concentration × 100.

### Strain isolation and identification
For strain isolation, microbial culture in the microfluidic chips was collected after 12, 36, and 96 h growth and plated on ISEM agar plate. After 5-day incubation at 25 °C, the single colonies were picked, precultivated in ISEM and then frozen in 25% glycerol at −80 °C. The isolated strains were identified via Sanger sequencing with universal primers 27F/1492R. A total of 764 bacterial strains were isolated from the microfluidic chips. These strains belonged to five different genera and covered core planktonic and biofilm members, including *Arthrobacter*, *Rhodococcus*, *Pseudomonas*, *Staphylococcus,* and *Pseudarthrobacter*.

### Pairwise interspecies interaction analysis
The bacterial isolates were precultivated in ISEM. After 48 h growth, the bacterial culture was washed with PBS buffer and resuspended to an OD$_{600}$ of 1.0. In pairwise co-culture interaction between *Arthrobacter* and other species, 200 μL of ISEM was inoculated with 1 μL of each culture. To evaluate the effect of exometabolites of *Arthrobacter* strains, bacterial isolates were cultivated in a conditioned medium consisting of the cell-free supernatant of *Arthrobacter* culture and fresh ISEM. The supernatant of *Arthrobacter*, named *Arthrobacter* spent medium, was prepared by cultivating *Arthrobacter* strains in ISEM for 48 h, followed by centrifugation and filtration[25]. The conditioned medium was prepared by mixing the spent medium with an equal volume of fresh ISEM. Each well of a 96-well microplate contained 200 μL conditioned medium which was inoculated with 2 μL of each isolate. The microplates were incubated at room temperature for 48 h. The biofilms in the microplates were stained with 1% crystal violet, solubilized with 95% ethanol and quantified by measuring the absorbance at 590 nm.

### Genus-specific qPCR assay
The bacterial populations in monoculture and co-culture were quantified by qPCR with genus-specific primers. The bacterial culture was

prepared and cultivated as described above for the pairwise interaction analysis. The initial cell concentrations of the identical strain in the monoculture and different co-culture combinations were the same. After 24 h growth, the planktonic cells and biofilms attached on the inner surface of the microplates were collected at the exponential phase to evaluate the relative fitness of different strains[66]. Total cell numbers were calculated as the sum of biofilm and planktonic cells. DNA was extracted using TIANamp Bacteria DNA Kit (TIANGEN). Four independent biological replicates were carried out. The PCR program was as follows: 95 °C for 10 min, 40 cycles of 95 °C for 15 s and 60 °C for 60 s, followed by 15 s at 95 °C and 60 s at 60 °C. The primer sets were ART-F (GGGGACATTCCACGTTT) and ART-R (GCACCTGTTTCCAGG CG) for *Arthrobacter* strains, PSE435F (ACTTTAAGTTGGGAGGAAGGG) and PSE686R (ACACAGGAAATTCCACCACCC) for *Pseudomonas* strains and Rho627F (ATTCCGTGGAAGGAACCCAC) and Rho885R (TCGCGTCGTTTGTGAAAACC) for *Rhodococcus* strains. The specificity of PCR products was verified via gel electrophoresis and melting curve analysis (Supplementary Figs. 32 and 33). As described above, target DNA fragments were cloned into pMD8-T vector to develop the standard curves (Supplementary Fig. 33). The amplification efficiencies calculated from the corresponding standard curves were 94.6% for *Arthrobacter*, 94.8% for *Pseudomonas* and 101.5% for *Rhodococcus*. To estimate the absolute cell number, the linearity between PCR amplification and colony-forming unit numbers for each strain was determined (Supplementary Fig. 34).

### Classification of bacterial interactions

In co-culture interaction, the minimum ($Y_{min}$), average ($Y_{ave}$), maximum ($Y_{max}$), and sum ($Y_{sum}$) biofilm yield of each isolate were determined based on the biofilm grown in monoculture[67]. The co-culture interaction was defined as positive, when the biofilm yield of co-culture ($Y_{co}$) was higher than $Y_{sum}$ ($Y_{co} > Y_{sum}$). The relationship was considered as negative if the co-culture biofilm was less than or equal to the sum of the monocultures ($Y_{sum} \geq Y_{co}$). The co-culture interaction was strong negative when $Y_{co}$ was less than $Y_{min}$, while a weak negative relationship was determined when $Y_{sum} \geq Y_{co} \geq Y_{min}$. To assess the effect of extracellular metabolites in social interaction, the biofilm or planktonic growth in conditioned medium ($Y_c$) was compared with that in unconditioned medium ($Y_u$)[25]. A higher biofilm or planktonic growth in conditioned medium ($Y_c \geq Y_u$) indicated a positive interaction. The interaction was classified as weak negative when $1 > Y_c/Y_u \geq 0.5$. A substantial reduction in biofilm or planktonic growth ($Y_c/Y_u < 0.5$) was an indication of strong negative.

### Genome sequencing, assembly, and annotation

Total DNA of ASV2 *A. ramosus*, ASV1 *P. fluorescens* and ASV11 *R. erythropolis* was extracted using the cetyltrimethylammonium bromide (CTAB) method[68]. Whole genome sequencing was performed on the Illumina NovaSeq PE150 platform. Quality filtering and adaptor removal were carried out using AdapterRemoval (version 2.2.2)[69]. The filtered reads were assembled via A5-MiSeq (version 20160825)[70] and SPAdes (version 3.12.0)[71]. The draft genomes were annotated on the Integrated Microbial Genomes Expert Review (IMG ER) platform. The genomic data is available in the IMG database with the IMG Genome IDs of 2934219947, 2934206586 and 2934877363.

### RNA-seq analysis

ASV1 *P. fluorescens* and ASV11 *R. erythropolis* were grown in ISEM as monoculture and in co-culture with ASV2 *A. ramosus*. After 4 h growth, the bacterial culture was harvested at the exponential phase (Supplementary Fig. 35). Total RNA was extracted using TRIzol extraction method[72]. Paired-end sequencing was performed on the Illumina NovaSeq PE150 platform. Htseq-count (version 1.6.0) was used to generate the read count of individual gene and the differentially expressed genes (fold change ≥2, p-value < 0.05) were identified by

DESeq2 (version 1.28.1)[73–75]. KEGG functional enrichment analysis was performed using ClusterProfiler (version 4.0.0)[76].

### Chrome azurol S (CAS) assay

The siderophore production by *Pseudomonas* and *Rhodococcus* strains was detected by the CAS assay[77,78]. Bacterial cells were cultivated in ISEM for 48 h. 100 μL of filtered supernatant was mixed with an equal volume of CAS assay solution (1 mM CAS, 0.1 mM $FeCl_3$). The mixture was allowed to stand for 20 min and then the absorbance was measured at 630 nm. Enterobactin, a bacterial siderophore, was used to construct the standard curve (Supplementary Fig. 36).

### Construction of siderophore synthesis mutant

The role of siderophore in microbial interaction was elucidated by deleting the siderophore synthetase gene (*sfnaD*, locus tag: Ga0508010_06_50115_52037) in ASV1 *P. fluorescens* using homologous recombination. The mutant strain was constructed following a previously developed procedure[79,80]. The upstream and downstream flanking regions of *sfnaD* were amplified with the primer pairs sidupS/sidupA and siddwS/siddwA, respectively. The PCR products were ligated into the suicide plasmid pDS3.0 to generate plasmid pDS3.0-ΔsfnaD. After verifying the plasmid by sequencing, pDS3.0-ΔsfnaD was electroporated into ASV1 *P. fluorescens* and selected on LB plates containing 30 μg/mL gentamicin. The strains with pDS3.0-ΔsfnaD was then cultivated in LB for 48 h and then spread on LB plates with 20% sucrose to screen sucrose-positive and gentamicin-negative phenotypes (*sfnaD* deletion mutant). The deletion of *sfnaD* was verified by PCR amplification and sequencing (Supplementary Fig. 37). The primer sets used in this study are shown in Supplementary Table 9. The siderophore production of ΔsfnaD in ISEM was below detection limit (0.045 mM), which was consistent with the reduced halo size of the ΔsfnaD colony on CAS agar plate (Supplementary Fig. 38).

### Flavin identification and quantification

FAD, FMN and RF were identified and quantified via high-resolution LC−QTOF (1260−6540, Agilent) equipped with a ZORBAX SB-C18 column (2.1 × 150 mm, 5 μm)[81,82]. A mobile-phase gradient was used with methanol containing 0.5% acetic acid as mobile phase A and 0.5% acetic acid (pH 4.5) as mobile phase B. The solvent gradient conditions were applied at a flow rate of 0.2 mL/min as follows: mobile phase A increased from 7 to 100% over the first 7 min, held until 9 min, then returned to 7% at 14 min and equilibrated for 4 min. Flavin standards were purchased from Shyuanye Co. (China) for RF and FAD and Bidepharm Co. (China) for FMN. To confirm the excretion of flavins, ASV1 *P. fluorescens* was cultivated in ISEM and the extracellular metabolites were extracted using the same approach as for the exometabolome analysis. The LC−QTOF was operated in positive ion mode. The MS parameters were set as follows: gas temperature at 350 °C, nebulizer gas at 40 psi, capillary voltage at 4000 V, and the fragmentor at 170 V. Flavins was identified by matching the accurate mass and retention time of authentic standards (Supplementary Fig. 39). Flavins were quantified with a diode array detector at 267 nm. Calibration curves were prepared in triplicates (Supplementary Fig. 40). The method precision and accuracy were assessed at three concentration levels (Supplementary Table 10).

### Reporting summary

Further information on research design is available in the Nature Portfolio Reporting Summary linked to this article.

## Data availability

Source data are provided with this paper. The raw amplicon and transcriptomic sequencing data are deposited in the NCBI SRA

database with the accession number of PRJNA764456 and PRJNA813193. The reference libraries for metabolite identification are available from METLIN (http://metlin.scripps.edu), MassBank (https://massbank.eu/MassBank/), LipidMaps (https://www.lipidmaps.org/), and mzCloud (https://www.mzcloud.org/). The taxonomy reference is available at the Greengenes database (http://greengenes.lbl.gov). The KEGG pathways can be accessed through the KEGG database (https://www.genome.jp/kegg/). The genomic data for bacterial isolates is available in the IMG database (https://img.jgi.doe.gov/). Raw data including metabolomics data, confocal images of biofilm during colonization, dynamics of community composition, detected metabolites in metabolomics, differentially expressed genes in transcriptomic data, biofilm yield and planktonic growth in monoculture and co-culture, abundances of different genotypes in co-culture and concentrations of AAs during biofilm development have been deposited in the figshare database (https://figshare.com/projects/Cooperative_interactions_drive_spatial_segregation/156834). Source data are provided with this paper.

## Code availability

The computer code used for image processing and null model analysis have been stored in the figshare database (https://figshare.com/projects/Cooperative_interactions_drive_spatial_segregation/156834).

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

## Acknowledgements

Y.W. and P.C. acknowledge funding from the National Key Research Program of China (2020YFC1806802), the National Natural Science Foundation of China (42225706, 42177281, 42177283), Royal Society Newton Advance Fellowship (NAF/R1/191017) and Fundamental Research Funds for the Central Universities (2662022ZHYJ001, 2662021JC012, 2662023PY010). C.L.P. acknowledges funding from the European Research Council (ERC) under the European Union's Horizon 2020 research and innovation programme (Grant agreement No. 725613 MinOrg). K.X. is funded by the Hundred Talents Program of the Chinese Academy of Sciences. We thank Assoc. Prof. Yujie Xiao and Meina He for their invaluable assistance in mutant construction. We would also like to thank Prof. Feng Ju and Prof. Shuo Jiao for helpful comments and suggestions.

## Author contributions

Y.W. and P.C. conceived and designed the project. Y.W. and C.F. performed the experiments and analyzed the data. P.S. and Z.L. fabricated the microfluidic chips. Y.W. and C.F. interpreted the data in discussion with P.C. and C.L.P. Y.W. and C.F. wrote the manuscript with input from all authors.

## Competing interests

The authors declare no competing interests.
