## [Peer Review File · Nature Communications]

Point-by-Point Response to Reviewer's Comments and Criticisms

REVIEWER COMMENTS

Reviewer #1 (Remarks to the Author):

In this manuscript, Wu et al. use a customised microfluidics method to investigate the formation of biofilms in complex multispecies communities of bacteria. They identify a free-living bacteria (*Arthrobacter*) which helps initiate biofilm formation by the hydrolysis of D-amino acids. The work is of importance to a number of fields including; biofilm researchers and microbial ecologists. Given the importance of biofilms in pathogenicity it will likely be of interest to a wide audience and could have significant implications. The finding that free-living bacteria helps stimulate biofilm growth in other species is interesting and relatively novel. The methodology used encompasses a range of methods that have been conducted well, but the sample number of n=3-4 microfluidic chips is low. However, there are a few interpretations in the authors analysis which I feel are not supported by their results.

1. My main concern about this manuscript is the authors conclusion that the stimulation of biofilm growth by *Arthrobacter* through the hydrolysis of DAA is an example of mutualism. This is a very significant claim that is not thoroughly discussed. Examples of cross-species mutualism are relatively rare and of significant importance in ecology. By suggesting that the interaction is one of cooperation, it implies that the trait (DAA metabolism) has evolved to benefit other non-self bacteria. The authors do not test if *Arthrobacter* gains a benefit from the consumption of DAA, if it does then it may not be an example of cooperation, *Arthrobacter* is simply metabolising a nutrient and did not evolve to cooperate with any other bacteria (for example the relationship between elephant and dung beetle, the elephant dung benefits the dung beetle but the elephant didn't evolve to cooperate).

Response: We agree with the reviewer that the cooperation between *Arthrobacter* and biofilm-forming species wasn't demonstrated in the previous manuscript. To address this concern, the influence of DAA on the growth of *Arthrobacter* and the fitness effect exerted by the biofilm-

forming species were evaluated. The presence of DAA didn't significantly influence the growth of *Arthrobacter* strains (Supplementary Fig. 15), while the growths of *Arthrobacter* in 42 of 50 coculture combinations were enhanced compared to those in monoculture (Supplementary Fig. 4 & 5). Based on the CAS assay, siderophore was detected in the culture supernatant of *Pseudomonas* and *Rhodococcus* strains (Supplementary Fig. 9), confirming the production of public goods. The media conditioned by the biofilm-forming species also stimulated the growth of *Arthrobacter* (Supplementary Fig. 10). Therefore, *Arthrobacter* enhances biofilm formation via DAA hydrolysis and the benefited biofilm-forming species secrete public goods siderophore to provide a collective benefit. With the return benefits from biofilm-forming species, *Arthrobacter* evolved to hydrolyze DAA cooperatively. The mutually beneficial interaction between biofilm-forming species and *Arthrobacter* can be classed as cooperation. Following the insightful comments, we have added the following sentences (P14, L276-284).

“Biofilm-deficient or more planktonic *Arthrobacter* triggers biofilm formation via removal of biofilm inhibitors. Although *Arthrobacter* doesn't gain fitness benefits through DAA hydrolysis (Supplementary Fig. 15), it receives return benefits from biofilm-forming species which secrete public goods. The mutually beneficial interaction facilitates the occupation of segregated niches by free-living *Arthrobacter* and biofilm-forming bacteria and shapes community spatial organization. It indicates that the planktonic microbial minorities can impose strong selection pressure on biofilm community succession, by serving as inhibitor scavengers and, in turn, enhance their own fitness in the pore fluid.”

2. Secondly, the authors state that this is an example of mutualism. Mutualism requires that *Arthrobacter* receives a benefit from its interacting partner/s. The authors do not show this. While the relative abundance of *Arthrobacter* increases in the planktonic environment this isn't sufficient to suggest that *Arthrobacter* is receiving a benefit from biofilm associated bacteria. The authors do report an upregulation of public goods from *P. fluorescens* but do not provide evidence that these increase the fitness of *Arthrobacter* or can be utilised by *Arthrobacter*.

Response: We agree and have evaluated the secretion of siderophore via the CAS assay. *Pseudomonas* and *Rhodococcus* strains were found to produce substantially more siderophore than *Arthrobacter* strains (Supplementary Fig. 9). To assess whether biofilm-forming species benefit *Arthrobacter*, *Arthrobacter* strains were cultivated in the media conditioned by *Pseudomonas* and *Rhodococcus* strains. Compared with the fresh ISEM, the conditioned media significantly enhanced the growth of *Arthrobacter* strains (Supplementary Fig. 10). Therefore, the biofilm-forming species can provide a benefit to *Arthrobacter* via public goods production. In response to the comments, we have added the following sentences in the text (**P11, L216-222**).

“On the other hand, siderophore which acts as public goods was detected in the supernatants of *Pseudomonas* and *Rhodococcus* cultures (Supplementary Fig. 9). Compared with ISEM, the media conditioned by ASV1 *P. fluorescens* and ASV11 *R. erythropolis* significantly enhanced the growth of most *Arthrobacter* strains (Supplementary Fig. 10). These results support our hypothesis that excreted metabolites of free-living *Arthrobacter* can induce spatial segregation by promoting biofilm formation, which simultaneously enhances the fitness of *Arthrobacter* in planktonic communities via public goods production.”

3. There are other concerns about this paper. Though the authors do demonstrate that DAAs can inhibit biofilm formation, and that *Arthrobacter* is capable of hydrolysing DAA, the authors should conclusively demonstrate that there is no competition between *Arthrobacter* and biofilm forming bacteria as competition has been known to induce biofilm production in *Pseudomonads* (Oliveira, 2015).

Response: We have quantified the abundance of *Arthrobacter* and biofilm-forming species in the co-culture using qPCR. The relative frequencies of *Arthrobacter* strains after 24 h growth were comparable to the initial proportion which suggested that *Arthrobacter* and biofilm forming species didn't exclude each other in co-culture (Supplementary Fig. 4 & 5). Moreover, the media conditioned by the biofilm-forming species was found to significantly enhance the growth of *Arthrobacter* (Supplementary Fig. 10), indicating that *Arthrobacter* receive benefits from biofilm-forming species. As a response, the following sentences have been added for clarification (**P9, L173-**

L185).

“The abundance of different strains in co-culture systems was quantified via qPCR to evaluate the individual fitness. The total cell numbers of *Arthrobacter* and biofilm-forming strains in 42 and 37 of 50 co-culture mixtures were significantly higher than those in monoculture, respectively (Supplementary Fig. 4 & 5). Consistent with the observed community spatial structure in microfluidic chambers (Fig. 2E), the co-culture biofilm was mainly composed of the biofilm-forming species (median frequency of biofilm-forming strains in biofilm = 0.646, Wilcoxon-signed rank test versus 0.5, $n = 50$ combinations, $p < 0.01$), and the plankton was dominated by *Arthrobacter* (median frequency of *Arthrobacter* strains in plankton = 0.878, Wilcoxon-signed rank test versus 0.5, $n = 50$ combinations, $p < 0.01$). After 24 h growth, the frequencies of *Arthrobacter* in all the 50 co-culture combinations were comparable to the initial proportions (Wilcoxon-signed rank test versus *Arthrobacter* frequency in inoculum, $n = 4$ replicates for each combination, $p > 0.05$), which suggested that *Arthrobacter* and biofilm-forming species didn't exclude each other in co-culture.”

Minor issues:

The layout of Figure 2 is confusing. The panel indicating ASV taxon relationships could be included as a separate panel or included within Panel C.

Response: We agree and have replotted the figure. The taxonomy legend has been positioned on the right side. The following sentences were added in the figure caption to make it clearer (**P34, L759-760 & P34, L765-766**).

“Line colors correspond to different ASVs in the taxonomy legend at right.”

“The relative contribution of different genera to the community succession. Columns are colored based on the taxonomy (see the taxonomy legend on the right).”

The authors do not give enough description to allow their results to be replicated. Software and

parameters used for the metabolite identification should be included. iCAMP has many different parameters used in phylogenetic binning, if these are kept as default it should still be included in the manuscript.

Response: The parameters and settings for the metabolite identification and iCAMP analysis have been added to the method section (P21, L427-433 & P22, L443-451). We have also included the R and MATLAB scripts with the revised manuscript (P27, L547-552).

“Metabolite identification. The raw MS data were converted to mzXML format using MSConvert and were subsequently processed by XCMS for peak picking and alignment ¹. Metabolite identification was performed by accurate mass and MS/MS matching against METLIN, MassBank, LipidMaps, and mzCloud reference libraries ^{2,3,4}. The XCMS settings were as follows: method = "centWave", ppm = 15, peakwidth = c(10,60), mzwid = 0.025, minfrac = 0.5, and bw = 5. Only metabolites at level 2 identification (putatively annotated compounds) were used for downstream statistical analysis.”

“The analysis was conducted using the parameters recommended by the developers ⁵. The threshold of phylogenetic distance was set to 0.2 and the minimal bin size was 12. The 11,985 observed ASVs were classified into 442 different phylogenetic bins. The null model distribution was generated using 1000 randomizations ⁶. The null model algorithm “taxa shuffle” was applied to shuffle the taxa across the phylogenetic tree and randomize the phylogenetic relationship within phylogenetic bins ⁷. The β -Net Relatedness Index (β NRI) and taxonomic β -diversities using modified Raup–Crick metric (RC) were determined to identify the ecological process governing each bin. The cutoffs of significant β NRI value, significant RC value and significant one-side confidence level were 1.96, 0.95 and 0.975, respectively.”

“Source data including confocal images of biofilm during colonization, dynamics of community composition, detected metabolites in metabolomics, differentially expressed genes in transcriptomic data, abundances of different genotypes in co-culture and concentrations of AAs during biofilm development and computer code used for image processing and null model analysis

have been deposited in the figshare database

(https://figshare.com/projects/Cooperative_interactions_drive_spatial_segregation/156834).”

REFERENCE

1. Chambers MC, et al. A cross-platform toolkit for mass spectrometry and proteomics. *Nat. Biotechnol.* **30**, 918-920 (2012).
2. Horai H, et al. MassBank: a public repository for sharing mass spectral data for life sciences. *J Mass Spectrom* **45**, 703-714 (2010).
3. Sud M, et al. Lmsd: Lipid maps structure database. *Nucleic Acids Res.* **35**, D527-D532 (2007).
4. Wang J, Peake DA, Mistrik R, Huang Y. A platform to identify endogenous metabolites using a novel high performance Orbitrap MS and the mzCloud Library. *Blood* **4**, 2-8 (2013).
5. Ning D, et al. A quantitative framework reveals ecological drivers of grassland microbial community assembly in response to warming. *Nat. Commun.* **11**, 1-12 (2020).
6. Kembel SW, et al. Picante: R tools for integrating phylogenies and ecology. *Bioinformatics* **26**, 1463-1464 (2010).
7. Kembel SW. Disentangling niche and neutral influences on community assembly: assessing the performance of community phylogenetic structure tests. *Ecol. Lett.* **12**, 949-960 (2009).

The authors switch between X-h and X h when reporting time measurements.

Response: We have unified the description of time measurements as “X h”.

Line 104 references (Fig. 2B & sss2)

Response: The typo has been corrected.

Numbers of differentially expressed genes are reported for *A. ramosus* but not for *P. fluorescens* or *R. erythropolis* (Line 183).

Response: The numbers of differentially expressed genes for *P. fluorescens* and *R. erythropolis*

have been added in the text (P10, L188-189).

“Coculture with *A. ramosus* induced the differential expression of 2,165 and 161 genes in *P. fluorescens* and *R. erythropolis*, respectively.”

To conclude, the manuscript is well written and multispecies communities are of great interest, however, I feel the authors have made claims not supported by their data or have not discussed how their data support the claims they have made.

Reviewer #2 (Remarks to the Author):

This article investigates interactions between different genotypes within a natural consortium of soil bacteria. The authors find that one species of bacteria degrades the compounds that inhibit biofilm formation of other species, which ultimately allow the latter to become numerically dominant in the mixed species biofilm community. A very diverse array of analytical techniques are employed, along with diversity of different sophisticated data analysis tools – it is clear that a great deal of time and thought were required to put this story together. I find the basic premise of the story compelling and it is impressive that the authors were able to demonstrate a clear mechanism that drives succession in this natural community and isolate the compounds responsible. However, I do have several questions about the methods used and the interpretations of the results.

Major comments:

- In the title and throughout the manuscript it is noted that the two focal genotypes are “spatially segregated”, because one ultimately forms a biofilm, whilst the other remains in the plankton phase. Clearly these different genotypes occupy different niches, but they are not spatially segregated in the usual way, as they can still be in (more or less) direct contact with one another. I think it would be more appropriate to suggest that these two focal genotypes instead “occupy different niches”?

Response: The review raised an important issue. We agree with the reviewer that the segregated genotypes in soil and other natural environments are found to distribute in disconnected habitats which prevent interaction. Meanwhile, the spatial segregation was also observed within a local microbial community^{8,9}. For example, cell expansion in multi-strain biofilms can induce population subdivision into monoclonal sectors even when the different strains are initially well-mixed^{10, 11, 12}. Compared with biofilms containing mixed populations, the spatial segregated structure increases the frequency of interaction between cells of the same genotype and favors the cooperative behaviors^{13, 14}. Our study demonstrated that *Arthrobacter* and *Pseudomonas* cells were segregated from the mixed planktonic inoculum to occupy the ambient fluid and grain surface. In the different niches, the interacting partners are almost the neighboring cells with the same genotype. Although free-living cells can contact biofilms in the fluid, the spatial structuring of these two genotypes in

the porous environments modifies their respective interaction neighborhoods. Therefore, the spatial arrangement of free-living and biofilm cells can be recognized as spatial segregation. In response to the reviewer's comment, we have reworded the following sentence to make it clearer (**P8, L144-147**).

“The emergence of patches dominated by one genus represents the occurrence of spatial segregation in the early stage of biofilm development, through which free-living *Arthrobacter* and biofilm-forming *Pseudomonas* occupied the ambient fluid and grain surface, respectively and structured their local neighborhoods^{9, 10, 11, 12}.”

REFERENCE

8. Nadell CD, Drescher K, Foster KR. Spatial structure, cooperation and competition in biofilms. *Nat. Rev. Microbiol.* **14**, 589-600 (2016).
9. Eigentler L, Kalamara M, Ball G, MacPhee CE, Stanley-Wall NR, Davidson FA. Founder cell configuration drives competitive outcome within colony biofilms. *ISME J* **16**, 1512-1522 (2022).
10. Bottery MJ, Passaris I, Dytham C, Wood AJ, van der Woude MW. Spatial organization of expanding bacterial colonies is affected by contact-dependent growth inhibition. *Curr. Biol.* **29**, 3622-3634. e3625 (2019).
11. Frost I, et al. Cooperation, competition and antibiotic resistance in bacterial colonies. *ISME J* **12**, 1582-1593 (2018).
12. Sharma A, Wood KB. Spatial segregation and cooperation in radially expanding microbial colonies under antibiotic stress. *ISME J* **15**, 3019-3033 (2021).
13. Nadell CD, Bucci V, Drescher K, Levin SA, Bassler BL, Xavier JB. Cutting through the complexity of cell collectives. *Proc. Royal Soc. B* **280**, 20122770 (2013).
14. Dragoš A, et al. Division of labor during biofilm matrix production. *Curr. Biol.* **28**, 1903-1913. e1905 (2018).

- The number of cells in the planktonic phase and the number of cells in the biofilm phase are frequently compared with each other (e.g. in line 99). However, it appears from the methods that

fluid is continually being flowed through the device, such that planktonic cells have a limited residence time in the actual microfluidic device. How then does one directly measure the “planktonic fraction” and ultimately compare these two populations?

Response: As the planktonic community was continuously flushed out from microfluidic chips, we collected the effluent of microfluidic chips to quantify the cell density of planktonic communities. The biofilm density was then determined as the difference between the total and planktonic cell densities. We have added details to the method section (**P18, L359-368**).

“The effluent of microfluidic chips was collected to determine the cell density of planktonic communities. Thereafter, all bacterial cells including biofilms and planktonic cells were vigorously flushed out from the microfluidic chips with PBS buffer. The cell densities of planktonic and the total microbial communities in the microfluidic chips were quantified by qPCR using the primer pair Eub338F/Eub518R. To construct the standard curve for qPCR, the PCR product from the model organism *P. putida* KT2440 was cloned into pMD8-T vector and electroporated into *E. coli* DH5 α competent cells. Serial dilutions of the plasmid DNA were then amplified using the SYBR Green PCR Supermix (BioRad). The amplification efficiency determined based on the standard curve was 104.7%. The specificity of the amplified product was confirmed by melt curve analysis and gel electrophoresis (Supplementary Fig. 17 & 18).

It would be very surprising if the number of planktonic cells within the microfluidic device at any given time was ever on par with the number of biofilm cells, as the latter has a much higher cell density per unit volume. Can you more clearly articulate how you compare these two different populations and what assumptions are being made?

Response: The biofilm and planktonic cell densities in Figure 1D represented the number of bacterial cells per unit volume of the microfluidic chamber. As the microfluidic chips were inoculated with the soil bacterial suspension, the densities of biofilm and planktonic cells were comparable in the early stage of colonization. After the formation of mature biofilm (>72 h), the biofilm cell density in microfluidic chips were two orders of magnitude higher than that of the

planktonic community (Fig. 1D). More details of the quantification procedure have been provided and the following sentence has been added in the figure caption to make it clearer (P19, L368-373 & P35, L842-843).

“The amount of biofilm cells was calculated as the difference between the total and planktonic cell densities:

$$D_{biofilm} = (C_{entire} - D_{planktonic} \times V_{chip})/V_{chip}$$

where $D_{biofilm}$ and $D_{planktonic}$ are the cell densities of biofilm and planktonic communities in the microfluidic chamber, C_{entire} is the total cell number in the PBS eluate, V_{chip} is the volume of the microfluidic chamber (3.0 μ L).”

“The cell densities represented as the number of cells per unit volume of the microfluidic chamber.”

- Related to the above comment, given the constant flow condition it would seem any population dynamics in the planktonic phase would primarily occur in the tube of media connected to the microfluidic device, rather than within the microfluidic device itself. It would be good for the authors to better explain their microfluidic setup and make it clear the effect that this might have on the interpretation of their results.

Response: In the experimental setup, the fresh medium was injected into the microfluidic chamber at a constant flow rate to sustain microbial growth. The effluent containing planktonic cells and exometabolites was continuously discharged from the microfluidic chips. The continuous flow system enabled real-time monitoring of planktonic community and exometabolite dynamics. Based on the reviewer’s suggestion, we added more details about the microfluidic setup (P18, L350-355).

“The microfluidic chip comprised a flow channel with one inlet and one outlet. The inlet tubing connected the microfluidic chip to a syringe which was mounted on a syringe pump to feed fresh ISEM for microbial growth. The effluent containing free living (planktonic) cells and exometabolites was collected at the outlet. The continuous flow system provided a well-defined and constant condition for biofilm development and enabled the continuous monitoring of planktonic

community and exometabolite dynamics.”

- The authors claim to have found a “cooperative” interaction between different bacterial genotypes. While the degradation of biofilm inhibitors by *Arthrobacter* is demonstrated to increase the amount of biofilm produced by *Pseudomonas*, the benefit to *Arthrobacter* is not actually demonstrated. Rather they find that the expression of putative “public goods” are upregulated in *Pseudomonas*, but the fitness benefit to *Arthrobacter* is speculative. (Note: the pairwise competitions do not resolve the final amount of each genotype) Thus, I do not think “cooperation” has actually been demonstrated. Moreover, can the overall fitness benefit to *Pseudomonas* actually be demonstrated, rather than just increased biofilm formation? As noted above, it is not clear whether the planktonic+biofilm measurements are indicative of an overall fitness benefit, given the concern noted above.

Response: We agree that the benefit to *Arthrobacter* wasn’t demonstrated in the previous version of our manuscript. In response to the reviewer’s comments, we confirmed the secretion of siderophore by the biofilm-forming species via CAS assay (Supplementary Fig. 9). *Arthrobacter* strains were cultivated in the media conditioned by *Pseudomonas* and *Rhodococcus* to assess the influence of secreted metabolites from biofilm-forming species. Compared with the fresh ISEM, the conditioned media significantly enhanced the growth of *Arthrobacter* strains (Supplementary Fig. 10). It indicated that *Arthrobacter* can receive return benefits from biofilm-forming species. The relative fitness of *Pseudomonas* in plankton and biofilm were calculated based on amplicon sequencing and FISH analysis respectively. As each method may introduce distinct bias to the fitness indices and the relative fitness is not critical for our conclusion, we have removed this discussion on the relative fitness in plankton and biofilm from the revised manuscript. The overall fitness of *Pseudomonas* can be assessed based on their abundance in the total community including biofilm and plankton (Fig. 2B). The microbial growth in the microfluidic chip and increased frequency of *Pseudomonas* suggested that the overall fitness of *Pseudomonas* in the porous medium increased during biofilm colonization. We also characterized the total fitness in the pairwise interaction experiments by calculating the sum of biofilm and planktonic cells. The total fitness of *Pseudomonas* in 21 of 25 co-culture combinations were significantly higher than those in monoculture. Following the

insightful comment, we have added the following sentences in the text (P11, L216-222 & P9, L173-185).

“On the other hand, siderophore which acts as public goods was detected in the supernatants of *Pseudomonas* and *Rhodococcus* cultures (Supplementary Fig. 9). Compared with ISEM, the media conditioned by ASV1 *P. fluorescens* and ASV11 *R. erythropolis* significantly enhanced the growth of most *Arthrobacter* strains (Supplementary Fig. 10). These results support our hypothesis that excreted metabolites of free-living *Arthrobacter* can induce spatial segregation by promoting biofilm formation, which simultaneously enhances the fitness of *Arthrobacter* in planktonic communities via public goods production.”

“The abundance of different strains in co-culture systems was quantified via qPCR to evaluate the individual fitness. The total cell numbers of *Arthrobacter* and biofilm-forming strains in 42 and 37 of 50 co-culture mixtures were significantly higher than those in monoculture, respectively (Supplementary Fig. 4 & 5). Consistent with the observed community spatial structure in microfluidic chambers (Fig. 2E), the co-culture biofilm was mainly composed of the biofilm-forming species (median frequency of biofilm-forming strains in biofilm = 0.646, Wilcoxon-signed rank test versus 0.5, $n = 50$ combinations, $p < 0.01$), and the plankton was dominated by *Arthrobacter* (median frequency of *Arthrobacter* strains in plankton = 0.878, Wilcoxon-signed rank test versus 0.5, $n = 50$ combinations, $p < 0.01$). After 24 h growth, the frequencies of *Arthrobacter* in all the 50 co-culture combinations were comparable to the initial proportions (Wilcoxon-signed rank test versus *Arthrobacter* frequency in inoculum, $n = 4$ replicates for each combination, $p > 0.05$), which suggested that *Arthrobacter* and biofilm-forming species didn't exclude each other in co-culture.”

- Line 258: I don't think your description of references 29 and 30 is accurate. Do these really investigate “free living” cells (i.e. planktonic cells)?

Response: We agree and have replaced the references with the correct ones. The previous studies demonstrated that biofilms outcompeted the planktonic cells by positioning themselves into the

more nutrient-rich region^{15,16}.

REFERENCE

15. Rainey PB, Rainey K. Evolution of cooperation and conflict in experimental bacterial populations. *Nature* **425**, 72-74 (2003).

16. Madsen JS, et al. Facultative control of matrix production optimizes competitive fitness in *Pseudomonas aeruginosa PA14* biofilm models. *Appl. Environ. Microbiol.* **81**, 8414-8426 (2015).

Minor comments:

- Line 104: typo in figure call out.

Response: The typo has been corrected.

- A more thorough description of the iCAMP method would be useful.

Response: In response to the comment, we have added the following sentences to elaborate iCAMP analysis before introducing the results (**P7, L113-121**).

“The relative importance of different taxonomic groups to the total community succession was inferred via a phylogenetic-bin-based null model analysis (iCAMP)⁵. This framework divides observed ASVs into different taxonomic groups based on the phylogenetic relationships. The relative importance of ecological processes governing the turnovers of each group is determined by the null model analysis based on beta Net Relatedness Index (β NRI) and modified Raup-Crick metric (RC). The pairwise comparison with β NRI < -1.96 is an indicative of homogeneous selection, whereas that with β NRI > 1.96 interpreted as heterogenous selection. The taxonomic dissimilarity metric RC is applied for the pairwise comparisons with $|\beta$ NRI| \leq 1.96. RC values > 0.95 or < -0.95 represent homogenizing dispersal or dispersal limitation, while $|RC| \leq 0.95$ is interpreted as drift.”

REFERENCE

5. Ning D, et al. A quantitative framework reveals ecological drivers of grassland microbial community assembly in response to warming. *Nat. Commun.* **11**, 1-12 (2020).

- Line 132: Not clear what you meant by “heterogeneous”

Response: Based on the FISH analysis and 16S amplicon sequencing, *Pseudomonas* became more abundant in biofilm while its relative abundance in planktonic communities reduced significantly. It suggested that *Pseudomonas* cells were not uniformly distributed in the porous environment. We have reworded the following sentence for clarification (**P8, L139-141**).

“The opposite changes in the relative abundance of *Pseudomonas* in biofilm and planktonic communities suggests that *Pseudomonas* underwent a transition from planktonic to biofilm-forming in the early stage of biofilm development.”

- Line 163: Twenty and two ==> 22

Response: The suggested change has been made.

- Line 206: need better descriptions than “accumulating”, “declining”, and “invariant”. i.e. with respect to what?

Response: The measured exometabolites were grouped based on *k*-means clustering analysis. The cluster 1 (accumulating) and cluster 2 (declining) include exometabolites whose concentrations increased and decreased during biofilm development, respectively. The remaining exometabolites were grouped into cluster 3 (invariant). To address this concern, these three clusters have been designated as “released”, “consumed” and “the others”. The following sentence has been reworded for clarification (**P12, L231-233**).

“All the measured exometabolites were divided into three clusters based on their dynamics during biofilm development, including released (cluster 1), consumed (cluster 2) and the others (cluster 3)

(Supplementary Fig. 11-14).”

- Need to define “ISEM”, “NMDS” and other acronyms when they are first used.

Response: The full forms of these acronyms have been provided.

- Line 307: final not finial

Response: The typo has been corrected.

- Line 327: does the addition of any of these compounds cause biofilm detachment prior to imaging?

Response: The first step of FISH is to preserve the biofilm architecture using the cross-linking fixative formaldehyde. We have imaged the biofilm morphologies before and after the sample pretreatment for FISH analysis. The images have demonstrated that the pretreatment wouldn't induce biofilm detachment (Supplementary Fig. 20). As a response, we have added the following sentence (**P20, L388-389**).

“The formaldehyde fixation preserves biofilm integrity for subsequent sample pretreatment (Supplementary Fig. 20).”

- Supplementary Fig. 6: are these imaged at the same time into the experiment? Can you say when in the caption?

Response: Yes, all the biofilms were imaged after 48 h growth. We added the following sentence in the figure caption for clarification (**Supplementary Fig. 6**).

“Biofilm cells are cultured in microfluidic chips for 48 hours and then stained with DAPI (blue).”

Reviewer #3 (Remarks to the Author):

This manuscript used a combination of sequencing, transcriptomics, metabolomics, and imaging approaches to track the planktonic vs biofilm communities and metabolite dynamics in a microfluidic device representing porous environments. The authors showed that planktonic *Arthrobacter* promoted biofilm microbes, in particular *Pseudomonas* and *Rhodococcus*, by scavenging biofilm inhibitor D-amino acids (DAAs), thereby concluding cooperative interactions drove spatial segregation.

I appreciate the authors using metabolomic, transcriptomic, pairwise interaction analyses to explore the potential mechanism, including follow-up experiments to directly show the inhibitory effects of DAAs on biofilm formation, yet the latter is already a known phenomenon in the literature as also referred to by the authors.

One major component missing in the current study is how the biofilm-forming microbes in turn affect the free-living microbes, although the authors emphasized there were “mutualistic cooperation.” How would *Arthrobacter* grow in *Pseudomonas* conditioned medium? From the current results, there weren't many genes significantly altered in *Arthrobacter* when in co-culture. Further, *Arthrobacter* decreased substantially in the planktonic community overtime and DAAs increased at later stages as well. What caused these?

Response: The reviewer raises a very good point. The production of siderophore, which serves as cooperative public goods in microbial interactions, was detected in the culture supernatant of biofilm-forming species (Supplementary Fig. 9). When cultivating *Arthrobacter* strains in the media conditioned by *Pseudomonas* and *Rhodococcus*, the growth of *Arthrobacter* strains was significantly enhanced compare to those in fresh ISEM (Supplementary Fig. 10), indicating that biofilm-forming species provides fitness benefits to *Arthrobacter*. As the microfluidic devices was operated under continuous flow, the planktonic cells incapable of biofilm formation were continuously flushed out from the porous environment. Therefore, the reduced frequency of *Arthrobacter* in the planktonic community was attributed to its deficiency in biofilm formation and

failure in colonization under continuous flow. With the extinction of *Arthrobacter* in the porous environments, the metabolic capabilities of the microbial community towards DAA were diminished and DAA concentrations gradually increased to the original levels in ISEM. Following this insightful comment, we have added the following sentences in the discussion (P11, L216-222 & P15, L300-305).

“On the other hand, siderophore which acts as public goods was detected in the supernatants of *Pseudomonas* and *Rhodococcus* cultures (Supplementary Fig. 9). Compared with ISEM, the media conditioned by ASV1 *P. fluorescens* and ASV11 *R. erythropolis* significantly enhanced the growth of most *Arthrobacter* strains (Supplementary Fig. 10). These results support our hypothesis that excreted metabolites of free-living *Arthrobacter* can induce spatial segregation by promoting biofilm formation, which simultaneously enhances the fitness of *Arthrobacter* in planktonic communities via public goods production.”

“The changes in DAAs during biofilm colonization reveal a V-shaped pattern. At the late stage, the frequency of *Arthrobacter* capable of efficiently hydrolyzing DAAs decreased significantly due to its deficiency in biofilm formation. The extinction of *Arthrobacter* in the porous environments led to the diminished metabolic capabilities of the remaining microbial community towards DAAs, the concentration of which in the effluent gradually increased to the original level in ISEM (Fig. 4D).”

I know the authors applied many different methods in this work, yet still details for several of the experiments and analyses need to be added. For example, for the untargeted metabolomics analysis, under the continuous flushing condition, how much effluent samples were collected at each timepoint for metabolomics? How were the samples treated? What were the QCs and standards? How was the identification performed? Metabolite identification results should be provided as a supporting table. Similarly, amino acid identification and quantification need additional details including QA/QC and standard curves. Also, how did the continuous flushing with fresh medium affect the results, e.g., due to dilution?

Response: In response to the comment, we have added more details to the method section. For the

metabolomic analysis, we collected 100 μL of effluent from the microfluidic chip. Six independent biological replicates were sampled at each time point. The effluent samples were filtered and then mixed with a methanol/acetonitrile solution. The supernatant was lyophilized and resuspended in an acetonitrile/water mixture for LC-MS/MS-based metabolomic analysis. QC samples were prepared by pooling an identical volume of each sample. Five QC samples were injected prior to the analysis and after every ten samples to assess the platform stability. The metabolite identification was performed using XCMS. The data processing workflow and XCMS settings have now been added to the text. We also provide the metabolite identification results with the revised manuscript. For the amino acid identification and quantification by A300 amino acid analyzer, a certified standard amino acid solution containing 17 proteinogenic amino acids was used to calibrate the system and was injected at the beginning of the analysis and after every 10 runs. Regarding the quantification of amino acid enantiomers, QC samples of different amino acids at low, medium, and high concentrations were prepared to evaluate the accuracy and precision of the method (Supplementary Table 1). The standard curves of each amino acid have been included in the supporting information (Supplementary Fig. 24). The continuous-flow microfluidic system sustains biofilm growth under a steady-state condition. Moreover, the setup enabled real-time monitoring of planktonic community and exometabolite dynamics by sampling and analysis of the effluents. The following sentences have been added to make it clearer (P21, L410-418; P21, L427-433; P23, L455-459; P23, L470-475; P18, L350-355).

“100 μL of effluent was collected from each of six independent microfluidic chips at each time point. The effluent was filtered through a 0.2 μm microcentrifuge PVDF filter to remove bacterial cells. The filtrate was vortexed with 400 μL pre-cooled methanol/acetonitrile (50/50 v/v), left at $-20\text{ }^{\circ}\text{C}$ for 30 min and then centrifuged at 14,000 g for 20 min. The supernatant was lyophilized and resuspended in 100 μL of 50:50 acetonitrile/water solution. After centrifuged at 14,000 g for 15 min, the supernatant was harvested for LC-MS/MS measurement. Pooled QC samples were prepared by mixing an equal aliquot of effluent samples. Five aliquots of QC samples were injected prior to the analysis and after every ten runs to assess the analytical variance (Supplementary Fig. 21 & 22).”

“**Metabolite identification.** The raw MS data were converted to mzXML format using MSConvert

and were subsequently processed by XCMS for peak picking and alignment ¹. Metabolite identification was performed by accurate mass and MS/MS matching against METLIN, MassBank, LipidMaps, and mzCloud reference libraries ^{2,3,4}. The XCMS settings were as follows: method = "centWave", ppm = 15, peakwidth = c(10,60), mzwid = 0.025, minfrac = 0.5, and bw = 5. Only metabolites at level 2 identification (putatively annotated compounds) were used for downstream statistical analysis."

"The bacterial culture and the effluent were filtered through 0.2 µm filters prior to analysis. The total free AA concentration was determined via A300 amino acid analyzer (membraPure). The system was calibrated with a certified standard amino acid mixture of 17 proteinogenic amino acids (GBW(E)100062). The standard mixture was injected at the beginning of the analysis and after every 10 samples to evaluate the stability of the analytical platform."

"Calibration curves were constructed for each AA and their enantiomers (Supplementary Fig. 24). QC samples were prepared at 1.0, 4.0 and 9.0 mg/L to assess the precision and accuracy of the method (Supplementary Table 1). Three replicates of each QC level were analyzed. The relative standard deviation (RSD, %) was calculated as standard deviation/mean × 100. The accuracy was expressed as bias (%), which was calculated as (measured concentration - theoretical concentration)/theoretical concentration × 100."

"The microfluidic chip comprised a flow channel with one inlet and one outlet. The inlet tubing connected the microfluidic chip to a syringe which was mounted on a syringe pump to feed fresh ISEM for microbial growth. The effluent containing free living (planktonic) cells and exometabolites was collected at the outlet. The continuous flow system provided a well-defined and constant condition for biofilm development and enabled the continuous monitoring of planktonic community and exometabolite dynamics."

REFERENCE

1. Chambers MC, et al. A cross-platform toolkit for mass spectrometry and proteomics. *Nat. Biotechnol.* **30**, 918-920 (2012).

2. Horai H, et al. MassBank: a public repository for sharing mass spectral data for life sciences. *J Mass Spectrom* **45**, 703-714 (2010).
3. Sud M, et al. Lmsd: Lipid maps structure database. *Nucleic Acids Res.* **35**, D527-D532 (2007).
4. Wang J, Peake DA, Mistrik R, Huang Y. A platform to identify endogenous metabolites using a novel high performance Orbitrap MS and the mzCloud Library. *Blood* **4**, 2-8 (2013).

Additional details are also necessary to clarify image analysis, iCAMP, and community analysis (i.e., of planktonic vs biofilm vs mixture).

Response: According the reviewer's comment, we have added more details to the method section (**P19, L377-378; P18, L359-361; P19, L368-373; P22, L443-451**). The scripts for image and community analyses have also been provided with the revised manuscript (**P27, L547-552**).

“After image binarization, the biofilm thickness on each grain was computed as the average radial distance from points on the biofilm edges to the grain surface.”

“The effluent of microfluidic chips was collected to determine the cell density of planktonic communities. Thereafter, all bacterial cells including biofilms and planktonic cells were vigorously flushed out from the microfluidic chips with PBS buffer.”

“The amount of biofilm cells was calculated as the difference between the total and planktonic cell densities:

$$D_{biofilm} = (C_{entire} - D_{planktonic} \times V_{chip}) / V_{chip}$$

where $D_{biofilm}$ and $D_{planktonic}$ are the cell densities of biofilm and planktonic communities in the microfluidic chamber, C_{entire} is the total cell number in the PBS eluate, V_{chip} is the volume of the microfluidic chamber (3.0 μ L).”

“The analysis was conducted using the parameters recommended by the developers⁵. The threshold of phylogenetic distance was set to 0.2 and the minimal bin size was 12. The 11,985 observed ASVs

were classified into 442 different phylogenetic bins. The null model distribution was generated using 1000 randomizations ⁶. The null model algorithm “taxa shuffle” was applied to shuffle the taxa across the phylogenetic tree and randomize the phylogenetic relationship within phylogenetic bins ⁷. The β -Net Relatedness Index (β NRI) and taxonomic β -diversities using modified Raup–Crick metric (RC) were determined to identify the ecological process governing each bin. The cutoffs of significant β NRI value, significant RC value and significant one-side confidence level were 1.96, 0.95 and 0.975, respectively.”

“Source data including confocal images of biofilm during colonization, dynamics of community composition, detected metabolites in metabolomics, differentially expressed genes in transcriptomic data, abundances of different genotypes in co-culture and concentrations of AAs during biofilm development and computer code used for image processing and null model analysis have been deposited in the figshare database (https://figshare.com/projects/Cooperative_interactions_drive_spatial_segregation/156834).”

REFERENCE

5. Ning D, et al. A quantitative framework reveals ecological drivers of grassland microbial community assembly in response to warming. *Nat. Commun.* **11**, 1-12 (2020).
6. Kembel SW, et al. Picante: R tools for integrating phylogenies and ecology. *Bioinformatics* **26**, 1463-1464 (2010).
7. Kembel SW. Disentangling niche and neutral influences on community assembly: assessing the performance of community phylogenetic structure tests. *Ecol. Lett.* **12**, 949-960 (2009).

Why were *Staphylococcus* assessed (in Fig 3 and 5), which didn't seem to be a dominant genus in biofilm or planktonic communities?

Response: We agree with the suggestion and have excluded *Staphylococcus* in data analysis.

Per Nature guidelines, all individual replicate data points should be shown, not just an error bar in the bar chart, for example Fig 5A. Number of replicates should also be added to each figure, with

stats methods used for determining significance specified.

Response: The replicate data points are now shown in the figures. Number of replicates and the statistical methods for statistical significance testing have been added in the figure caption. For the figures that only illustrate the average and standard deviation, we have provided the source data file with the revised manuscript (P27, L547-552).

“Source data including confocal images of biofilm during colonization, dynamics of community composition, detected metabolites in metabolomics, differentially expressed genes in transcriptomic data, abundances of different genotypes in co-culture and concentrations of AAs during biofilm development and computer code used for image processing and null model analysis have been deposited in the figshare database (https://figshare.com/projects/Cooperative_interactions_drive_spatial_segregation/156834).”

The Discussion section can be enriched. Currently some part is not supported by the results. Please see below for detailed comments.

Line 93-95: How was image analysis performed? How was biofilm thickness measured? No info is provided in the Methods, Line 315. Pls make the code and images used for analysis available as supplemental.

Response: We performed image binarization and then measured biofilm thickness as the radial distance from biofilm edge to the grain surface. The following sentence has been added in the method section to make it clearer (P19, L377-378). The MATLAB scripts for image processing and original CLSM images have also been provided (P27, L547-552).

“After image binarization, the biofilm thickness on each grain was computed as the average radial distance from points on the biofilm edges to the grain surface.”

“Source data including confocal images of biofilm during colonization, dynamics of community composition, detected metabolites in metabolomics, differentially expressed genes in

transcriptomic data, abundances of different genotypes in co-culture and concentrations of AAs during biofilm development and computer code used for image processing and null model analysis have been deposited in the figshare database (https://figshare.com/projects/Cooperative_interactions_drive_spatial_segregation/156834).

Line 96-97: How were biofilm and planktonic communities collected? How was qPCR method validated? Info is not provided in the Methods, Line 310.

Response: The planktonic communities were collected at the outlet of microfluidic chips. Total bacterial cells in the microfluidic device including biofilms and planktonic cells were vigorously flushed out from the microfluidic chips with PBS buffer. The biofilm density was then determined as the difference between the total and planktonic cell densities. The gel electrophoresis, melting curve and standard curve analyses have been conducted to validate specificity and efficiency (Supplementary Fig. 17 & 18). In response to the comment, we have added the following sentences in text (**P18, L359-373**).

“The effluent of microfluidic chips was collected to determine the cell density of planktonic communities. Thereafter, all bacterial cells including biofilms and planktonic cells were vigorously flushed out from the microfluidic chips with PBS buffer. The cell densities of planktonic and the total microbial communities in the microfluidic chips were quantified by qPCR using the primer pair Eub338F/Eub518R. To construct the standard curve for qPCR, the PCR product from the model organism *P. putida* KT2440 was cloned into pMD8-T vector and electroporated into *E. coli* DH5 α competent cells. Serial dilutions of the plasmid DNA were then amplified using the SYBR Green PCR Supermix (BioRad). The amplification efficiency determined based on the standard curve was 104.7%. The specificity of the amplified product was confirmed by melt curve analysis and gel electrophoresis (Supplementary Fig. 17 & 18). The amount of biofilm cells was calculated as the difference between the total and planktonic cell densities:

$$D_{biofilm} = (C_{entire} - D_{planktonic} \times V_{chip})/V_{chip}$$

where $D_{biofilm}$ and $D_{planktonic}$ are the cell densities of biofilm and planktonic communities in the microfluidic chamber, C_{entire} is the total cell number in the PBS eluate, V_{chip} is the volume of the

microfluidic chamber (3.0 μ L).”

Since Methods is placed after Results, it would be helpful to add a brief description of what was done and why before diving into the results (e.g., Line 102).

Response: Following the reviewer’s suggestion, we have added the following sentence before introducing the results (**P6, L102-104**).

“To investigate the community succession during biofilm development, the total microbial community dynamics in the microfluidic chips were monitored by 16S amplicon sequencing.”

Line 102: Clarify if the results are of biofilm or free-living community, or mixture. In Methods, Line 342-343, both communities were sampled and analyzed, but here the results don’t seem to show both.

Response: The reviewer brings up a very good point that we overlooked in the previous version of our manuscript. The community analysis was based on the total microbial communities in the microfluidic chip including the biofilm and planktonic cells. As the total microbial communities were dominated by biofilm cells (Fig. 1D), we didn’t differentiate biofilms from the total microbial community. In response to the comment, we have reworded these sentences as follows (**P6, L104-105 & P20, L405-407**):

“The Bray-Curtis dissimilarity of total microbial communities between adjacent time intervals decreased as biofilm growth progressed (Fig. 2A).”

“At each sampling point, the microbial communities including biofilm and planktonic cells were collected from three independent microfluidic chips.”

Line 104: Pls correct “sss2?”

Response: The typo has been corrected.

Fig 2: Time 0 community composition is missing, so we don't really know how the starting point looked like. Fig 2C: are these abundances in the biofilm or total community?

Response: We have added the initial community structure in the Figure 2B. Figure 2B demonstrates the succession of total microbial communities including biofilm and planktonic cells in microfluidic chips. The following sentence has been added in the figure caption for clarification (P34, L758-759).

“B, The relative abundances of the TOP 20 ASVs (covering 86.3% of total reads) in the total microbial community.”

Line 105, Fig 2B: The decrease in early stage seems to be only from the 1st to the 2nd timepoint; with only 2 points, linear regression would not be an appropriate analysis here.

Response: We agree and have removed the linear regression analysis from the revised manuscript. The sentences have been rephrased as follows (P6, L106-108).

“During the initial stage, the community diversity and richness exhibited a significant decrease (Supplementary Fig. 2). Nominal changes were observed after the development of mature biofilms (>72 h) and the communities approached a steady state.”

Line 113-120, Fig 2D, 2E: Pls add more details explaining how the relative importance of these different processes was determined. It is unclear how taxonomic groups are linked to ecological processes. Also, are all the processes listed in Fig 2D relevant in biofilms? In text pls change “selection” to “homogeneous selection” to be consistent with Fig 2D and it may be helpful to also explain the difference between homogeneous vs heterogeneous selection.

Response: Based on the reviewer's comment, we have added more details to explain how the relative importance of different ecological processes was assessed (P7, L113-124). Fig. 2D demonstrated the assembly processes of the total microbial community in the microfluidic chips including biofilm

and planktonic cells. According to the comment, the “selection” has been changed to “homogeneous selection” (P7, L124). The following sentence has been reworded to explain homogeneous selection in community assembly (P7, L128-130).

“The relative importance of different taxonomic groups to the total community succession was inferred via a phylogenetic-bin-based null model analysis (iCAMP) ⁵. This framework divides observed ASVs into different taxonomic groups based on the phylogenetic relationships. The relative importance of ecological processes governing the turnovers of each group is determined by the null model analysis based on beta Net Relatedness Index (β NRI) and modified Raup-Crick metric (RC). The pairwise comparison with β NRI < -1.96 is an indicative of homogeneous selection, whereas that with β NRI > 1.96 interpreted as heterogenous selection. The taxonomic dissimilarity metric RC is applied for the pairwise comparisons with $|\beta$ NRI| \leq 1.96. RC values > 0.95 or < -0.95 represent homogenizing dispersal or dispersal limitation, while $|\text{RC}| \leq 0.95$ is interpreted as drift. The relative importance of individual processes is weighted by the relative abundance of each taxonomic groups and summed to estimate their relative importance in controlling community succession.”

“These results demonstrate that after a temporary stochastic period (≤ 12 h), the microbial communities were driven by homogeneous abiotic and biotic environmental conditions and *Pseudomonas* and *Arthrobacter* were the key taxa shaping community structure.”

REFERENCE

5. Ning D, et al. A quantitative framework reveals ecological drivers of grassland microbial community assembly in response to warming. *Nat. Commun.* **11**, 1-12 (2020).

Line 133: This is a bit confusing, since the relative fitness was calculated based on relative abundance, so this creates a circle in reasoning. Also, please explain why relative fitness was calculated as in Line 345.

Response: We agree with the reviewer that the relative fitness was determined on the basis of the

relative abundance data. The higher relative abundance is associated with the increased fitness. In response to this comment, the sentence regarding the relative fitness has been removed from the text and the following sentence has been reworded to make it clearer (P8, L139-141):

“The opposite changes in the relative abundance of *Pseudomonas* in biofilm and planktonic communities suggests that *Pseudomonas* underwent a transition from planktonic to biofilm-forming in the early stage of biofilm development.”

Line 156: Pls write out the full name of “ISEM” at its first appearance.

Response: The full name of ISEM has been added in the text (P9, L162-164).

“Although all the isolates showed substantial planktonic growth in the intensive soil extract medium (ISEM), *Arthrobacter* strains exhibited minimal biofilm forming capability (Fig. 3B).”

Fig 3B, 3D: What are those “unresolved” interactions? Why were *Staphylococcus* assessed, which didn’t show up as a dominant genus in biofilm communities? Also in Fig 5A. They are not among the top ASVs based on Fig 2C or in the planktonic communities (Sup Fig 3).

Response: The unresolved interaction represents the interactions which cannot be classified as cooperation or competition. We have changed “unresolved” to “neutral” in the revised manuscript and the following sentence has been rephrased (P25, L517-518). According to the reviewer’s comment, we have excluded *Staphylococcus* strain in the data analysis.

“The remaining relationship was designated as neutral.”

Sup Fig 5: caption for “NES” is confusing, “the members of the gene set are enriched at the top of the ordered transcriptomic data?” Also, the significance level needs to be added to the figure.

Response: GSEA algorithm ranks the genes based on the gene expression fold change in co-culture versus monoculture. The most significantly upregulated and downregulated genes in co-culture

were ranked at the top and bottom of the gene list, respectively. A significant positive NES for the KEGG pathway indicates that the genes in the pathway are overrepresented at the top of the ranked gene list. As a response, we have added the following sentences in the figure caption (**Supplementary Fig. 6**). The significance level is now shown in the figure.

“GSEA ranked genes based on the \log_2 fold change between co-culture and monoculture by which most significantly upregulated and downregulated genes in co-culture were at the top and bottom of the gene list, respectively. A significant positive normalized enrichment score (NES) represents that the genes in the KEGG pathway are enriched at the top of the ordered gene list, while negative NES indicates that the genes in the pathway are overrepresented at the bottom of the ranked list. The nominal P value for the observed NES was determined relative to null distribution. *, $p < 0.05$; **, $p < 0.01$; ***, $p < 0.001$.”

Line 184: Pls include a supplementary table or data file to list all the differentially expressed genes, including the fold changes and sig. levels, for all relevant results, e.g., Sup Fig 5, 7.

Response: The lists of differentially expressed genes have been provided as a Source Data file.

Sup Fig 8: Why for each cluster, there are two sets of dash lines, for 10 and 90%? Pls clarify.

Response: The two sets of dash lines represented 25th and 75th percentiles, and 10th and 90th percentiles. We have replotted the figure and the different percentiles are indicated by red and blue lines, respectively. The following sentence has been added in the figure caption for clarification (**Supplementary Fig. 11**):

“The blue lines represent 10th and 90th percentiles and the red lines indicate 25th and 75th percentiles.”

Sup Fig 8-11: How z-scores were calculated should be clarified. Fig 9-11: Should point out the ones that were significant. What do the labels of metabolites mean (e.g., “N001” and “P001”)?

Response: The z-scores were calculated using the peak area of each metabolite across all samples. The Spearman's correlation coefficient and statistical significance are now shown in the figure. "P" and "N" in the labels indicate the metabolites detected in positive and negative mode. The following sentences were added to address this concern (**Supplementary Fig. 11 & 12**):

"The abundance of each exometabolite (peak areas from LC-MS/MS) was z-score normalized across all samples."

"The first letter in the labels indicates the metabolite detected in positive (P) or negative (N) mode. Spearman correlation coefficients between metabolite abundance and the incubation period were determined. *, $p < 0.05$; **, $p < 0.01$; ***, $p < 0.001$."

Line 195: The conditioned medium experiments only showed *Pseudomonas* and *Rhodococcus* had higher growth, but don't provide evidence for whether these two can in turn enhance *Arthrobacter* fitness.

Response: The reviewer makes a good suggestion. We have cultivated the *Arthrobacter* strains in the media conditioned by ASV1 *P. fluorescens* and ASV11 *R. erythropolis*. Compared with ISEM, the conditioned media significantly enhanced the growth of *Arthrobacter* strains (**Supplementary Fig. 10**). We have added the following sentences in the text (**P11, L217-222**).

"Compared with ISEM, the media conditioned by ASV1 *P. fluorescens* and ASV11 *R. erythropolis* significantly enhanced the growth of most *Arthrobacter* strains (**Supplementary Fig. 10**). These results support our hypothesis that excreted metabolites of free-living *Arthrobacter* can induce spatial segregation by promoting biofilm formation, which simultaneously enhances the fitness of *Arthrobacter* in planktonic communities via public goods production."

Line 198: Here immediately after describing the interaction experiments, it would be helpful to clarify that metabolomics analysis was done for effluent from the microfluidic device.

Response: Based on the reviewer's suggestion, we have reworded the sentence as follows (**P12, L224-226**):

“Towards understanding exometabolite-driven spatial segregation, the dynamics of extracellular metabolites in the effluents of microfluidic chips were unraveled via untargeted metabolomics analysis.”

Line 203-213, Fig 4A, B: Need to clarify if community structure and individual OTU abundance are of biofilm or planktonic or mixture.

Response: As the exometabolites in the effluents of microfluidic chips were secreted by both biofilm and planktonic cells, the Procrustes and correlation analyses were based on total microbial communities. We have added details to the figure caption (**P38, L789-793**).

“**A**, Procrustes rotation of the total microbiota composition nonmetric multidimensional scaling (NMDS) plot with the metabolome NMDS plot (Procrustes correlation = 0.72, $p = 0.001$). **B**, Correlation between the ASVs (relative abundance > 0.1% across total microbial communities in microfluidic chips) and amino acids.”

Fig 5A: Why were these 5 DAAs chosen? What do the bar charts on the right represent?

Response: Totally six DAAs were detected during biofilm development (Fig. 4D). We mistakenly omitted D-leucine in our previous manuscript. The bar charts in Fig. 5A represent the total residual DAA concentration in the bacterial culture. We have replotted the figure and included the concentration of residual D-leucine. The data points of three independent replicates are now shown in the Fig. 5A. The following sentence has been added in the figure caption for clarification (**P39, L803**).

“The bar chart shows the total residual DAAs in the culture.”

Line 263: The secretion of “public goods in return” by biofilm species is not examined in the experiments.

Response: The reviewer makes a very important point. We have evaluated the production of siderophore via the chrome azurol S (CAS) assay and determined the growth of *Arthrobacter* in the media conditioned by biofilm-forming isolates. The results clearly showed the siderophore production by *Pseudomonas* and *Rhodococcus* strains (Supplementary Fig. 9). The media conditioned by *Pseudomonas* and *Rhodococcus* strains were also demonstrated to enhance the growth of *Arthrobacter* (Supplementary Fig. 10). It indicated that the biofilm-forming species can provide a benefit to the planktonic cells via the production of public goods. In response to the comment, we have added the following sentences in the text (P11, L216-222).

“On the other hand, siderophore which acts as public goods was detected in the supernatants of *Pseudomonas* and *Rhodococcus* cultures (Supplementary Fig. 9). Compared with ISEM, the media conditioned by ASV1 *P. fluorescens* and ASV11 *R. erythropolis* significantly enhanced the growth of most *Arthrobacter* strains (Supplementary Fig. 10). These results support our hypothesis that excreted metabolites of free-living *Arthrobacter* can induce spatial segregation by promoting biofilm formation, which simultaneously enhances the fitness of *Arthrobacter* in planktonic communities via public goods production.”

Line 271-276: This part of discussion is not supported by data. There are no experiments showing the mechanism of DAA increase during later stage, e.g., who produced DAAs. Additional experiments of isolates may be helpful to support this.

Response: We agree and have removed the discussion on the potential role of *Pseudomonas* strains in DAA synthesis. The paragraph has been rephrased as follows (P15, L300-310):

“Our results reveal that DAAs are key metabolites to mediate spatial segregation. The changes in DAAs during biofilm colonization reveal a V-shaped pattern. At the late stage, the frequency of *Arthrobacter* capable of efficiently hydrolyzing DAAs decreased significantly due to its deficiency

in biofilm formation. The extinction of *Arthrobacter* in the porous environments led to the diminished metabolic capabilities of the remaining microbial community towards DAAs, the concentration of which in the effluent gradually increased to the original level in ISEM (Fig. 4D). As a common group of secondary metabolites and major constituents of bacterial walls, these amino acid enantiomers can be actively excreted and passively released to surrounding environments^{18,19}. Micromoles of DAA per gram dry weight, which is sufficient to suppress biofilm formation at micromolar levels^{20,21}, were detected in sediment and soil^{18,22,23}. DAAs may therefore play a pervasive role in governing biofilm community structure over a wide range of ecosystems.”

REFERENCE

18. Hu Y, Zheng Q, Zhang S, Noll L, Wanek W. Significant release and microbial utilization of amino sugars and D-amino acid enantiomers from microbial cell wall decomposition in soils. *Soil Biol. Biochem.* **123**, 115-125 (2018).
19. Aliashkevich A, Alvarez L, Cava F. New Insights Into the Mechanisms and Biological Roles of D-Amino Acids in Complex Eco-Systems. *Front. Microbiol.* **9**, 683 (2018).
20. Kolodkin-Gal I, Romero D, Cao S, Clardy J, Kolter R, Losick R. D-amino acids trigger biofilm disassembly. *Science* **328**, 627-629 (2010).
21. Leiman SA, May JM, Lebar MD, Kahne D, Kolter R, Losick R. D-amino acids indirectly inhibit biofilm formation in *Bacillus subtilis* by interfering with protein synthesis. *J. Bacteriol.* **195**, 5391-5395 (2013).
22. Langerhuus AT, *et al.* Endospore abundance and D: L-amino acid modeling of bacterial turnover in holocene marine sediment (Aarhus Bay). *Geochim. Cosmochim. Acta* **99**, 87-99 (2012).
23. Vranova V, *et al.* The significance of D-amino acids in soil, fate and utilization by microbes and plants: review and identification of knowledge gaps. *Plant Soil* **354**, 21-39 (2012).

Methods

Line 291-299: Where was the dry soil used for ISEM sampled? And how variable was the composition batch to batch? Pls add the exact components and conc. of the added minerals, vitamins and amino acids.

Response: We used the same soil sample for soil microorganism extraction and ISEM preparation. The surface soil was collected from a paddy field of National Agro-Ecosystem Observation and Research Station (Jiangxi Province, China, 116°55'E, 28°15'N) (P17, L321-323). To evaluate the variability of ISEM, we compared the composition of ISEM prepared from three independent batches via three-dimensional excitation–emission matrix fluorescence spectroscopy and UV-vis. The spectroscopic analyses revealed a similar chemical composition of ISEM from different batches (Supplementary Fig. 16). In response to the reviewer's concern, we have added the following sentences in the text. The final concentrations of minerals and micronutrients in ISEM have been added to the method section (P17, L331-340 & P18, L341-343).

“Each liter of ISEM consisted of 0.2 L soil extract solution, 0.23 g KH_2PO_4 , 0.23 g K_2HPO_4 , 0.23 g $\text{MgSO}_4 \cdot 7\text{H}_2\text{O}$, 0.33 g NH_4NO_3 , 0.25 g NaHCO_3 , 5 mg of each D-amino acids (D-valine, D-methionine, D-leucine, D-phenylalanine, D-threonine, and D-tryptophan), 1 mL of the vitamin stock solution (thiamine hydrochloride, 0.5 g/L; riboflavin, 0.5 g/L; niacin, 0.5 g/L; pyridoxine HCl, 0.5 g/L; inositol, 0.5 g/L; calcium pantothenate, 0.5 g/L; β -aminobenzoic acid, 0.5 g/L; biotin, 0.25 g/L), 2 mL of the selenite-tungstate solution (NaOH , 0.5 g/L; $\text{Na}_2\text{SeO}_3 \cdot 5\text{H}_2\text{O}$, 3 mg/L; $\text{Na}_2\text{WO}_4 \cdot 2\text{H}_2\text{O}$, 4 mg/L) and 2 mL of the trace element solution SL-10 (HCl , 2.8 g/L; $\text{FeCl}_2 \cdot 4\text{H}_2\text{O}$, 1.5 g/L; ZnCl_2 , 70 mg/L; $\text{MnCl}_2 \cdot 4\text{H}_2\text{O}$, 100 mg/L; H_3BO_3 , 6 mg/L; $\text{CoCl}_2 \cdot 6\text{H}_2\text{O}$, 190 mg/L; $\text{CuCl}_2 \cdot 2\text{H}_2\text{O}$, 2 mg/L; $\text{NiCl}_2 \cdot 6\text{H}_2\text{O}$, 24 mg/L; $\text{Na}_2\text{MoO}_4 \cdot 2\text{H}_2\text{O}$, 36 mg/L)^{24, 25}.”

“Three batches of ISEM were prepared to assess the batch-to-batch variations. On the basis of three-dimensional excitation–emission matrix (3D-EEM) and UV-vis spectroscopic analyses, the composition of ISEM was consistent across the independent batches (Supplementary Fig. 16).”

REFERENCE

24. Tschuch A, Pfennig N. Growth yield increase linked to caffeate reduction in *Acetobacterium woodii*. *Arch. Microbiol.* **137**, 163-167 (1984).
25. Widdel F, Kohring G-W, Mayer F. Studies on dissimilatory sulfate-reducing bacteria that decompose fatty acids. *Arch. Microbiol.* **134**, 286-294 (1983).

Line 348-349: In the above section, Line 342, it reads like biofilm communities were assessed using 16S, but why here FISH result instead of 16S was used?

Response: The relative fitness, calculated based on the relative abundance data, interprets the similar findings as the relative abundance. Both of them suggest the opposite changes of *Pseudomonas* abundance in biofilm and planktonic communities. As different methods may introduce distinct bias and the relative fitness is not critical for our conclusion, we have removed this section from the revised manuscript.

Line 350: More details are needed; for example, how much sample was collected under the continuous flushing conditions, how it was pre-treated before injecting into LCMS, what types of QCs were used, etc. Metabolite identification results should be provided as a supporting table. Data analysis methods are missing, including in reporting summary; for example, how identification was conducted, which standard library was used etc.?

Response: At each time point, 100 μ L of effluent was collected from each of six microfluidic chips. The effluent samples were filtered and then mixed with a methanol/acetonitrile solution. The supernatant was lyophilized and resuspended in an acetonitrile/water mixture for LC-MS/MS-based metabolomic analysis. Pooled QC samples were prepared to assess the platform stability. The list of identified exometabolites has been provided with the revised manuscript. The metabolite identification was performed using XCMS platform. The accurate mass and the fragmentation data of each metabolite peak were searched in METLIN, MassBank, LipidMaps and mzCloud databases. The data processing workflow and XCMS settings have now been added to the text (**P21, L410-418 & P21, L427-433**).

“100 μ L of effluent was collected from each of six independent microfluidic chips at each time point. The effluent was filtered through a 0.2 μ m microcentrifuge PVDF filter to remove bacterial cells. The filtrate was vortexed with 400 μ L pre-cooled methanol/acetonitrile (50/50 v/v), left at -20 $^{\circ}$ C for 30 min and then centrifuged at 14,000 g for 20 min. The supernatant was lyophilized and

resuspended in 100 μ L of 50:50 acetonitrile/water solution. After centrifuged at 14,000 g for 15 min, the supernatant was harvested for LC-MS/MS measurement. Pooled QC samples were prepared by mixing an equal aliquot of effluent samples. Five aliquots of QC samples were injected prior to the analysis and after every ten runs to assess the analytical variance (Supplementary Fig. 21 & 22).”

“**Metabolite identification.** The raw MS data were converted to mzXML format using MSConvert and were subsequently processed by XCMS for peak picking and alignment ¹. Metabolite identification was performed by accurate mass and MS/MS matching against METLIN, MassBank, LipidMaps, and mzCloud reference libraries ^{2,3,4}. The XCMS settings were as follows: method = "centWave", ppm = 15, peakwidth = c(10,60), mzwid = 0.025, minfrac = 0.5, and bw = 5. Only metabolites at level 2 identification (putatively annotated compounds) were used for downstream statistical analysis.”

REFERENCE

1. Chambers MC, *et al.* A cross-platform toolkit for mass spectrometry and proteomics. *Nat. Biotechnol.* **30**, 918-920 (2012).
2. Horai H, *et al.* MassBank: a public repository for sharing mass spectral data for life sciences. *J Mass Spectrom* **45**, 703-714 (2010).
3. Sud M, *et al.* Lmsd: Lipid maps structure database. *Nucleic Acids Res.* **35**, D527-D532 (2007).
4. Wang J, Peake DA, Mistrik R, Huang Y. A platform to identify endogenous metabolites using a novel high performance Orbitrap MS and the mzCloud Library. *Blood* **4**, 2-8 (2013).

Line 370-377: More details are needed.

Response: We have added more details about null model analysis to the method section (**P22, L443-451**). The R script for iCAMP analysis has also been provided with the revised manuscript (**P27, L547-552**).

“The analysis was conducted using the parameters recommended by the developers ⁵. The threshold of phylogenetic distance was set to 0.2 and the minimal bin size was 12. The 11,985 observed ASVs

were classified into 442 different phylogenetic bins. The null model distribution was generated using 1000 randomizations ⁶. The null model algorithm “taxa shuffle” was applied to shuffle the taxa across the phylogenetic tree and randomize the phylogenetic relationship within phylogenetic bins ⁷. The β -Net Relatedness Index (β NRI) and taxonomic β -diversities using modified Raup–Crick metric (RC) were determined to identify the ecological process governing each bin. The cutoffs of significant β NRI value, significant RC value and significant one-side confidence level were 1.96, 0.95 and 0.975, respectively.”

“Source data including confocal images of biofilm during colonization, dynamics of community composition, detected metabolites in metabolomics, differentially expressed genes in transcriptomic data, abundances of different genotypes in co-culture and concentrations of AAs during biofilm development and computer code used for image processing and null model analysis have been deposited in the figshare database (https://figshare.com/projects/Cooperative_interactions_drive_spatial_segregation/156834).”

REFERENCE

5. Ning D, *et al.* A quantitative framework reveals ecological drivers of grassland microbial community assembly in response to warming. *Nat. Commun.* **11**, 1-12 (2020).
6. Kembel SW, *et al.* Picante: R tools for integrating phylogenies and ecology. *Bioinformatics* **26**, 1463-1464 (2010).
7. Kembel SW. Disentangling niche and neutral influences on community assembly: assessing the performance of community phylogenetic structure tests. *Ecol. Lett.* **12**, 949-960 (2009).

Line 378: More details are needed, including quality controls and standard curves.

Response: For the amino acid identification and quantification, a certified standard amino acid mixture containing 17 proteinogenic amino acids was used to calibrate the amino acid analyzer and injected prior to the analysis and after every ten runs to monitor the system stability. With respect to the quantification of amino acid enantiomers, QC samples of each amino acids at low, medium,

and high concentrations were prepared to evaluate the accuracy and precision of the method. The standard curves of each amino acid enantiomer have been included in the supporting information. We have added details to the method section (**P23, L455-459 & P23, L470-475**).

“The bacterial culture and the effluent were filtered through 0.2 µm filters prior to analysis. The total free AA concentration was determined via A300 amino acid analyzer (membraPure). The system was calibrated with a certified standard amino acid mixture of 17 proteinogenic amino acids (GBW(E)100062). The standard mixture was injected at the beginning of the analysis and after every 10 samples to evaluate the stability of the analytical platform.”

“Calibration curves were constructed for each AA and their enantiomers (Supplementary Fig. 24). QC samples were prepared at 1.0, 4.0 and 9.0 mg/L to assess the precision and accuracy of the method (Supplementary Table 1). Three replicates of each QC level were analyzed. The relative standard deviation (RSD, %) was calculated as standard deviation/mean × 100. The accuracy was expressed as bias (%), which was calculated as (measured concentration - theoretical concentration)/theoretical concentration × 100.”

Line 429: Why was 4-hour chosen as the timepoint?

Response: As bacterial cells are physiologically stable during the exponential growth phase²⁶, we harvested the microbial cells in the exponential phase based on the growth profile. To address the reviewer’s concern, the following sentence has been added for clarification (**P26, L532-533**).

“After 4 h growth, the bacterial culture was harvested at the exponential phase (Supplementary Fig. 28).”

REFERENCE

26. Fishov I, Zaritsky A, Grover N. On microbial states of growth. *Mol. Microbiol.* **15**, 789-794 (1995).

Reviewer comments, second round

Reviewer #1 (Remarks to the Author):

The authors have made significant efforts to address the concerns of previous reviewers. The additional experiments provide strong evidence that the interaction is an example of mutualism even if the method of benefit from biofilm forming bacteria to *Arthrobacter* is not identified. These additions have resulted in a much stronger manuscript. However, there are still some minor concerns about the work which need to be address.

- Line 140: The authors draw conclusions from two separate relative abundance measurements to say that *Pseudomonas* underwent a transition between planktonic to biofilm. This conclusion cannot be justified using relative abundances.
- Line 147: The authors state that both *Pseudomonas* and *Arthrobacter* 'structured their local neighbourhoods' without clarifying what they mean. How is *Arthrobacter* structuring its environment if it is in the ambient fluid?
- Line 155: Figure 2C does not show anything relating to abundance.
- Line 170-172: The authors use a measurement of biofilm production to describe interaction type and strength, comparing the monoculture of each partner to the coculture of the pair. This is a common technique in bacterial ecology. However, they state that a measurement of coculture biofilm greater than the max of monoculture biofilm is a 'weak positive' interaction. This is not justified in the text and conceptually does not suggest a positive interaction, weak or otherwise. For example, if in monoculture bacteria A produces 5 units and bacteria B produces 3 units, then in coculture A produces 4 and B produces 2 to give a total of 6. This would be described as a weak positive interaction even though both bacteria are producing less than in monoculture. Coculture biofilm production greater than the sum of the monocultures (what the authors call a 'strong positive' interaction) is more typical of a positive interaction.
- Line 200: The authors state that GESA revealed similar transcriptional responses between *P. fluorescens* and *R. erythropolis*. Can the authors offer a statistic for this comparison? From looking at Supplementary Figure 6, the gene sets do not look to overlap well and even conflict between the two strains.
- Line 217, Line 222 and Line 279: The authors state that they identified siderophores in the supernatant of *Pseudomonas* and *Rhodococcus* 'siderophore which acts as public goods was detected in the supernatants...'. They then go on to say biofilm formation 'simultaneously enhances the fitness of *Arthrobacter* in planktonic communities via public good production' and that *Arthrobacter* 'receives return benefits from biofilm-forming species which secrete public goods'. These statements imply that biofilm producing bacteria are secreting public goods, likely siderophores, which benefit *Arthrobacter*. Whilst this is one possible explanation it isn't necessarily true. Firstly, implying siderophores are the mechanism should be avoided as there is no evidence that *Arthrobacter* can use these siderophores. Secondly, there is little direct evidence that biofilm producing bacteria are producing any public good which *Arthrobacter* can utilise. Biofilm bacteria could be removing toxins or altering environmental conditions such as pH. These possibilities are not discussed.
- The authors use FISH to measure the relative abundances of *Pseudomonas* and *Arthrobacter* in biofilm. The authors state that the probes were validated in Supplementary Figure 19. This image is of two labelled colonies. Probe validation should include statistics on the sensitivity and specificity of each probe. Differences in which could alter the relative abundance metric.
- Figure 1D, the legend does not describe what the two colors (blue and green) relate to.

Reviewer #2 (Remarks to the Author):

This manuscript has undergone substantial revision.

All three reviewers pointed out that the previous manuscript did not demonstrate “cooperation” and new experiments have been performed to address this. These new experiments indicate that the planktonic *Arthrobacter* strain often exhibits enhanced growth in the spent media of the biofilm-forming *P. fluorescens* and *R. erythropolis* strains (Supplementary Fig. 10). This enhanced growth is speculated to result from the siderophore secretions produced by the latter two biofilm strains (Supplementary Fig. 9). This speculation could potentially be tested using mutants that lack siderophore production, but the results of such an experiment might prove difficult to interpret (e.g. if siderophore production affects the growth rate of the biofilm forming species).

In addition, new experiments show that *Arthrobacter* does not exhibit increased growth as the concentration of DAA in the media increases, in an effort to demonstrate that *Arthrobacter* does not receive a direct fitness benefit from the hydrolysis of DAA (Supplementary Fig. 15). Again, these experiments are suggestive, but not conclusive. The lowest concentration of DAA in the experiments shown in Supplementary Fig. 15 is 15 mg/L – however, it seems possible that *Arthrobacter* might exhibit less growth if the concentration of DAA was smaller than 15 mg/L. i.e. ideally one would compare the growth of *Arthrobacter* in the presence of DAA and to that in the absence of DAA. However, it is not clear if such a comparison would be feasible. Would it be possible to reduce the DAA concentration to a negligible level to conclusively test if *Arthrobacter* can obtain a benefit from DAA? (e.g. using a different media?).

In the revised manuscript, the authors have also clarified how the biofilm and planktonic fractions were calculated, in addition to providing some additional details regarding the microfluidic experiments. However, a few questions about these experiments remain. In particular, the authors still have not addressed my previous comment about planktonic cells and the potential for contamination upstream and downstream of the porous media microfluidic device.

The device only has a single inlet and a single outlet. Were cells introduced into the device through the upstream tube? If so, then how did you prevent the tubing (whose volume is surely much larger than that of the device) from getting contaminated with bacteria? In practice, it is very difficult to avoid the contamination of tubing without switching tubes mid-experiment, which is technically difficult because it introduces air bubbles that detaches bacteria from surfaces. Can you please clarify how the devices were inoculated with bacteria and how the effluent was collected from the porous device without being “contaminated” by the tubing?

In the revised methods, it is noted that the volume of the device is 3 μL , whilst the flow rate used in the experiment is 0.5 μL per minute, indicating that the residence time of planktonic bacteria within the device is only 6 mins. How then should one interpret the population dynamics of the “planktonic” bacteria, whose doubling time is surely larger than 6 mins? e.g. are the results being influenced by what is happening in the tubing upstream/downstream of the porous media device? e.g. are the changes in the numbers of planktonic *Arthrobacter* cells strongly influenced by changes in their rate of detachment from surfaces of the microfluidic device, rather than by changes in their growth rate? This is not clear and should be addressed.

Overall, I feel like this paper does constitute a significant advance. However, questions do remain as to whether (a) cooperation has been definitively demonstrated and (b) the interpretations of the population dynamics of planktonic bacteria within the (unavoidably small) microfluidic device that is presented are valid. If the former cannot be demonstrated definitively, then the claim that cooperation has been observed should be toned back significantly. In addition, if the reported phenomena is being strongly influenced by the population dynamics within the tubing on either side of the microfluidic device, then that should be spelled out more clearly.

Minor comments:

Supplementary Fig. 10 - It is noted that the concentration of DAA in standard ISEM media is 44.1 mg/L. From the methods it appears that six different DAAs are included in the standard recipe for ISEM media. It would be informative to spell out how the different DAA concentrations reported in Supplementary Fig. 10 were prepared? e.g. do they contain all six different supplemented DAAs, but just at a lower concentrations than the standard media or ????. Please clarify.

In Line 359 it says that "all bacterial cells including biofilms and planktonic cells were vigorously flushed out from the microfluidic chips with PBS buffer." What do you mean by "vigorously"? e.g. what were the flow rates and timescales used? Please elaborate.

Fig. 1D: The legend indicating which symbols corresponds to "biofilm" and which corresponds to "planktonic" have been removed in this version of the manuscript. These need to be added back in.

Supplementary Fig. 15: The caption says that "Different letters indicate significant differences ($p < 0.05$, one-way ANOVA).", but it does not state what "a" and "b" actually mean.

Line 308-309: "micromolar" is repeated twice in this sentence. This passage needs to be reworded for clarity.

Reviewer #3 (Remarks to the Author):

I thank the authors for adding additional conditioned medium growth experiments, siderophore measurements, and more method details during the revision.

The biggest concern about this work, as pointed out by all 3 reviewers during the last review, was that the "cooperation" focus of the manuscript was not supported. Now that by showing increased growth of *Arthrobacter* in the conditioned media of *Pseudomonas* and *Rhodococcus* and higher amounts of siderophore in the conditioned media, the authors concluded that *Pseudomonas* and *Rhodococcus* benefited *Arthrobacter* by producing siderophore as a public good. However, I don't think the increased growth of *Arthrobacter* can be attributed to siderophore—which was not shown to directly improve growth here (esp. this was not a Fe-limiting condition) and siderophore is only one metabolite present in the complex mixture of conditioned media.

Furthermore, this still doesn't support the major conclusion that cooperative interaction mediated via extracellular metabolites caused the spatial segregation of free-living *Arthrobacter* and biofilm-forming bacteria. In particular, in addressing one previous comment, the authors added Time 0 inoculum community composition to Fig 2B, which shows *Pseudomonas* was the absolute dominant genus from the very beginning. So a plausible main mechanism could be as simple as that when the dominant *Pseudomonas* formed biofilms in ISEM (with ~44 mg/L DAA), the other dominant genus, biofilm-deficient *Arthrobacter*, was left behind with an increase in the relative abundance in the planktonic communities first, but then decreased to very low abundance under continuous flushing with fresh ISEM.

That said, I agree that the biofilm-promoting effects of *Arthrobacter*-degrading DAAs could still have contributed in situ, but likely to a less extent (<25%) considering the shorter time frame during early biofilm establishment (much <48h) and the overall decrease of DAAs (Fig 4D) and observed effects in pure culture (Fig 5B).

Metabolomics: Even though it's level2 identification, the identification results should be provided, to include each metabolite's m/z, MS/MS similarity score, etc. Also, deposit of metabolomics spectra in MetaboLights under no. MTBLS3742 cannot be found.

Double check CLSM images in Source Data File; 72 h used a different stain compared with the others (Line94).

Line99: It's confusing to compare the cell densities of the planktonic vs biofilm communities. I can

understand Equation in Line 370 at a single sampling timepoint, but it's confusing to think about it when considering the planktonic cells are continuously flushed out (and lost) during constant flow of fresh ISEM.

Also, "The amount of biofilm cells was considerably higher than that of the planktonic community" is not true in the early stage of incubation.

Fig 1D: need to clarify which symbols represent planktonic vs biofilm.

Line 173-185 and Supplementary Fig. 4-5 are a bit abrupt and confusing. Suggest to add some transition about how SFig4-5 were done and are related to Fig 3A-B, considering there are both monocultures and cocultures, and only contained part of the isolates in Fig3. Were SFig4-5 done in microfluidic devices? It's confusing why there are 3 different monoculture bars in the figures and how the inocula were constructed which seem to be of different compositions?

Line 216: I thank the authors for adding the new spent medium experiments. This would need to revise Fig 3A and possibly add a subplot similar to 3D, to show the bidirectional growth promoting effects. Abstract needs to be revised to include this if keeping the current title.

Line 278: Should add ISME without DAA addition as the control for comparison.

Source data for Fig 3B, 3D need to be provided.

Line 416: Please clarify "Pooled QC samples were prepared by mixing an equal aliquot of effluent samples." Also, please clarify what these QC are.

Line 432: What is the MS/MS matching score?

Line 438: z-score may not be the ideal way here to analyze the data. Supplementary Fig. 12-14 may be easier to understand if relative abundances are shown as changes compared with 0-h levels.

SFig13: It's hard to understand why cluster2 (consumed group) had metabolites positively correlated with time.

Line 472: I may have missed it somewhere, but couldn't find Supplementary Table 1.

Line 496: Why this was done for 24 h but the other experiments, such as the conditioned medium experiment was done for 48 h.

Point-by-Point Response to Reviewer's Comments and Criticisms

REVIEWER COMMENTS

Reviewer #1 (Remarks to the Author):

The authors have made significant efforts to address the concerns of previous reviewers. The additional experiments provide strong evidence that the interaction is an example of mutualism even if the method of benefit from biofilm forming bacteria to *Arthrobacter* is not identified. These additions have resulted in a much stronger manuscript. However, there are still some minor concerns about the work which need to be address.

- Line 140: The authors draw conclusions from two separate relative abundance measurements to say that *Pseudomonas* underwent a transition between planktonic to biofilm. This conclusion cannot be justified using relative abundances.

Response: We agree that the changes in the relative abundance of *Pseudomonas* between biofilm and planktonic communities are insufficient to demonstrate the lifestyle transition from planktonic to biofilm. To address this concern, we calculated the proportion of *Pseudomonas* in biofilm to the total *Pseudomonas* cell number in microfluidic chambers based on the qPCR and amplicon sequencing results (Fig. 1D, 2B & Supplementary Fig. 4). The result revealed that the proportion of *Pseudomonas* in biofilm increased from $19.5 \pm 12.7\%$ to $97.3 \pm 10.0\%$, indicating *Pseudomonas* transited from planktonic to biofilm mode of life. The following sentences have been added in the text (**P8, L146-150**):

“Based on qPCR and amplicon sequencing analyses (Fig. 1D, 2B & Supplementary Fig. 4), $19.5 \pm 12.7\%$ of *Pseudomonas* in the microfluidic chamber inhabited biofilms at 12 h. The ratio further increased to $97.3 \pm 10.0\%$ at 36 h. The increased ratio of sessile *Pseudomonas* cells suggests that *Pseudomonas* underwent a transition from planktonic to biofilm-forming in the early stage of biofilm development.”

- Line 147: The authors state that both *Pseudomonas* and *Arthrobacter* ‘structured their local neighbourhoods’ without clarifying what they mean. How is *Arthrobacter* structuring its environment if it is in the ambient fluid?

Response: The dominance of a single genus in biofilm increases the interaction frequency between cells with similar genotypes^{1,2,3,4}. Meanwhile, the structure of planktonic communities determines

microbial interaction and metabolic capability through the production and consumption of exometabolites ^{5,6}. Our result also demonstrated that *Arthrobacter* contributed to the reduced D-amino acid level in the fluid. We have added the following sentence for clarification (**P9, L156-158**).

“The dominance in biofilm increases the frequency of interactions between cells with similar genotypes ^{1,2,3,4}, while the prevalent plankton taxa can exert a strong impact on the exometabolite pool ^{5,6}.”

REFERENCE

1. Eigentler L, Kalamara M, Ball G, MacPhee CE, Stanley-Wall NR, Davidson FA. Founder cell configuration drives competitive outcome within colony biofilms. *ISME J* **16**, 1512-1522 (2022).
2. Sharma A, Wood KB. Spatial segregation and cooperation in radially expanding microbial colonies under antibiotic stress. *ISME J* **15**, 3019-3033 (2021).
3. Bottery MJ, Passaris I, Dytham C, Wood AJ, van der Woude MW. Spatial organization of expanding bacterial colonies is affected by contact-dependent growth inhibition. *Curr. Biol.* **29**, 3622-3634. e3625 (2019).
4. Frost I, *et al.* Cooperation, competition and antibiotic resistance in bacterial colonies. *ISME J* **12**, 1582-1593 (2018).
5. Gao C-H, Cao H, Cai P, Sørensen SJ. The initial inoculation ratio regulates bacterial coculture interactions and metabolic capacity. *ISME J* **15**, 29-40 (2021).
6. Dolinšek J, Ramoneda J, Johnson DR. Initial community composition determines the long-term dynamics of a microbial cross-feeding interaction by modulating niche availability. *ISME Commun.* **2**, 77 (2022).

- Line 155: Figure 2C does not show anything relating to abundance.

Response: The description refers to Fig. 2B which shows the simultaneous increase in the relative abundance of *Pseudomonas* and *Arthrobacter* in the total community. We have revised the sentence as follows (**P9, L165-166**):

“Given the simultaneous increase in relative abundance (Fig. 2B), we hypothesize that a potential positive interaction between *Pseudomonas* and *Arthrobacter* induces spatial segregation.”

- Line 170-172: The authors use a measurement of biofilm production to describe interaction type and strength, comparing the monoculture of each partner to the coculture of the pair. This is a common technique in bacterial ecology. However, they state that a measurement of coculture biofilm greater than the max of monoculture biofilm is a ‘weak positive’ interaction. This is not justified in the text and conceptually does not suggest a positive interaction, weak or otherwise. For

example, if in monoculture bacteria A produces 5 units and bacteria B produces 3 units, then in coculture A produces 4 and B produces 2 to give a total of 6. This would be described as a weak positive interaction even though both bacteria are producing less than in monoculture. Coculture biofilm production greater than the sum of the monocultures (what the authors call a ‘strong positive’ interaction) is more typical of a positive interaction.

Response: The reviewer raises a very importance point regarding the definition of positive interaction in biofilm formation. As the induction of overall biofilm formation relative to the maximum biofilm yield of monocultures may not be contributed by enhanced biofilm formation of both interaction partners, we only define positive interaction when the coculture biofilm yield is higher than the sum of monocultures. According to the reviewer’s comment, we have revised the criteria for different interaction patterns (P28, L584-588 & P41, L904-907 & P10, 182-183).

“The co-culture interaction was defined as positive, when the biofilm yield of co-culture (Y_{co}) was higher than or equal to Y_{sum} ($Y_{co} \geq Y_{sum}$). The relationship was considered as negative if the co-culture biofilm was less than the sum of the monocultures ($Y_{sum} > Y_{co}$). The co-culture interaction was strong negative when Y_{co} was less than Y_{min} , while a weak negative relationship was determined when $Y_{sum} > Y_{co} \geq Y_{min}$.”

“Based on the minimum (Y_{min}), average (Y_{ave}), maximum (Y_{max}) and sum (Y_{sum}) monoculture biofilm yield of each member in co-culture and the co-culture biofilm yield (Y_{co}), the pairwise interaction can be classified as positive ($Y_{co} \geq Y_{sum}$), strong negative ($Y_{co} < Y_{min}$) or weak negative ($Y_{sum} > Y_{co} \geq Y_{min}$).”

“20 out of 25 co-culture combinations between *Arthrobacter* and *Pseudomonas* were assigned as positive interactions ($Y_{co} \geq Y_{sum}$).”

- Line 200: The authors state that GESA revealed similar transcriptional responses between *P. fluorescens* and *R. erythropolis*. Can the authors offer a statistic for this comparison? From looking at Supplementary Figure 6, the gene sets do not look to overlap well and even conflict between the two strains.

Response: Both *P. fluorescens* and *R. erythropolis* upregulated metabolic pathways involved in amino acid metabolism and secondary metabolite biosynthesis. We have replotted the heatmap based on KEGG categories to provide a clearer representation (Supplementary Fig. 7). The following sentences have been added in the text for clarification (P11, L212-216).

“GESA analysis revealed that co-culture also upregulated amino acid metabolism and secondary

metabolite biosynthesis in *R. erythropolis* (Supplementary Fig. 7). Each of these two biofilm-forming strains activated 4 different amino acid metabolic pathways. A total of 237 and 263 genes involved in the biosynthesis of cofactors and secondary metabolites were positively enriched in *P. fluorescens* and *R. erythropolis*, respectively.”

- Line 217, Line 222 and Line 279: The authors state that they identified siderophores in the supernatant of *Pseudomonas* and *Rhodococcus* ‘siderophore which acts as public goods was detected in the supernatants...’. They then go on to say biofilm formation ‘simultaneously enhances the fitness of *Arthrobacter* in planktonic communities via public good production’ and that *Arthrobacter* ‘receives return benefits from biofilm-forming species which secrete public goods’. These statements imply that biofilm producing bacteria are secreting public goods, likely siderophores, which benefit *Arthrobacter*. Whilst this is one possible explanation it isn’t necessarily true. Firstly, implying siderophores are the mechanism should be avoided as there is no evidence that *Arthrobacter* can use these siderophores. Secondly, there is little direct evidence that biofilm producing bacteria are producing any public good which *Arthrobacter* can utilise. Biofilm bacteria could be removing toxins or altering environmental conditions such as pH. These possibilities are not discussed.

Response: We concur with the reviewer that there are other potential explanations for the enhanced growth of *Arthrobacter* in the conditioned soil extract medium. To address this concern, we have measured the pH of ISEM before and after cultivating *Pseudomonas* and *Rhodococcus* strains. No significant change in pH has been observed. The growth of *Arthrobacter* was also enhanced in M9 minimal medium conditioned by *Pseudomonas* and *Rhodococcus* (Supplementary Fig. 10B). It suggests that *Arthrobacter* strains benefited from the metabolites secreted by the biofilm-forming strains. Based on the transcriptomic analysis, the genes of ASV1 *P. fluorescens* responsible for the synthesis of public goods, including siderophore, FMN and FAD, were significantly upregulated in co-culture (Fig. 3B). The production of these public goods was confirmed through CAS assay and LC-MS analysis (Supplementary Fig. 11 & 12). To further evaluate whether *Arthrobacter* can exploit these public goods, we constructed the siderophore synthesis mutant and supplemented flavins in the culture media. Deletion of siderophore synthetase gene reduced the growth of most *Arthrobacter* strains in the spent medium (Supplementary Fig. 13), while the addition of flavins significantly increased the growth of *Arthrobacter* in both ISEM and M9 medium (Supplementary Fig. 14). These results demonstrated that biofilm-forming strains can enhance the fitness of *Arthrobacter* via public goods production. As a response, we have added the following sentences in the text (P12, L230-241):

“On the other hand, ISEM conditioned by ASV1 *P. fluorescens* and ASV11 *R. erythropolis* significantly enhanced the growth of most *Arthrobacter* strains (Fig. 3D & Supplementary Fig. 10A).

The enhanced growth was also observed in M9 minimal medium (Supplementary Fig. 10B), which suggested *Arthrobacter* strains benefited from the metabolites secreted by the biofilm-forming strains. As public goods such as siderophore, FMN and FAD were detected in the supernatant of ASV1 *P. fluorescens* grown in ISEM (Supplementary Fig. 11 & 12), we further investigated whether *Arthrobacter* exploited these public goods by constructing the siderophore synthesis mutant ($\Delta sfnaD$) and adding flavins to the culture media. Deletion of the siderophore synthetase gene reduced the growth of most *Arthrobacter* strains in the spent medium of ASV1 *P. fluorescens* (Supplementary Fig. 13). Meanwhile, the supplementation of flavins at the same level as the ISEM conditioned by ASV1 *P. fluorescens* significantly enhanced the growth of *Arthrobacter* in both ISEM and M9 medium (Supplementary Fig. 14).”

- The authors use FISH to measure the relative abundances of *Pseudomonas* and *Arthrobacter* in biofilm. The authors state that the probes were validated in Supplementary Figure 19. This image is of two labelled colonies. Probe validation should include statistics on the sensitivity and specificity of each probe. Differences in which could alter the relative abundance metric.

Response: We have evaluated the specificity and sensitivity of FISH using the specific, non-sense and universal probes. The high detection rate of positive samples and low false positive signal frequency indicate that the labelling efficiency for *Pseudomonas* and *Arthrobacter* is comparable and FISH is a reliable approach to quantify their abundance in biofilm (P22, L440-445).

“The probes were validated with pure cultures (Supplementary Fig. 25 & 26). The universal bacterial probe EUB338 (5'-GCT GCC TCC CGT AGG AGT-3') and non-sense probe (5'-ACT CCT ACG GGA GGC AGC-3') were used as the positive and negative control, respectively. The specificity and sensitivity of FISH were assessed by calculating the detection rate, which is the proportion of DAPI-stained cells detected by FISH. The high detection rates of positive samples and few nonspecific false positive signals indicate that the FISH analysis is a feasible and reliable approach for quantifying *Pseudomonas* and *Arthrobacter* (Supplementary Fig. 27).”

- Figure 1D, the legend does not describe what the two colors (blue and green) relate to.

Response: We have added the figure legend in the revised manuscript.

Reviewer #2 (Remarks to the Author):

This manuscript has undergone substantial revision.

All three reviewers pointed out that the previous manuscript did not demonstrate “cooperation” and new experiments have been performed to address this. These new experiments indicate that the planktonic *Arthrobacter* strain often exhibits enhanced growth in the spent media of the biofilm-forming *P. fluorescens* and *R. erythropolis* strains (Supplementary Fig. 10). This enhanced growth is speculated to result from the siderophore secretions produced by the latter two biofilm strains (Supplementary Fig. 9). This speculation could potentially be tested using mutants that lack siderophore production, but the results of such an experiment might prove difficult to interpret (e.g. if siderophore production affects the growth rate of the biofilm forming species).

Response: Following the reviewer’s suggestion, we constructed a siderophore synthesis mutant to evaluate the influence of siderophore on the growth of *Arthrobacter*. Deletion of the siderophore synthetase gene (*sfnaD*) in ASV1 *P. fluorescens* was found to reduce the growth of most *Arthrobacter* strains in the conditioned media (Supplementary Fig. 13). The result confirmed that *Arthrobacter* strains benefited from the siderophore secreted by the biofilm-forming strains. The following sentences have been added in the text (P12, L234-241).

“As public goods such as siderophore, FMN and FAD were detected in the supernatant of ASV1 *P. fluorescens* grown in ISEM (Supplementary Fig. 11 & 12), we further investigated whether *Arthrobacter* exploited these public goods by constructing the siderophore synthesis mutant ($\Delta sfnaD$) and adding flavins to the culture media. Deletion of the siderophore synthetase gene reduced the growth of most *Arthrobacter* strains in the spent medium of ASV1 *P. fluorescens* (Supplementary Fig. 13). Meanwhile, the supplementation of flavins at the same level as the ISEM conditioned by ASV1 *P. fluorescens* significantly enhanced the growth of *Arthrobacter* in both ISEM and M9 medium (Supplementary Fig. 14).”

In addition, new experiments show that *Arthrobacter* does not exhibit increased growth as the concentration of DAA in the media increases, in an effort to demonstrate that *Arthrobacter* does not receive a direct fitness benefit from the hydrolysis of DAA (Supplementary Fig. 15). Again, these experiments are suggestive, but not conclusive. The lowest concentration of DAA in the experiments shown in Supplementary Fig. 15 is 15 mg/L – however, it seems possible that *Arthrobacter* might exhibit less growth if the concentration of DAA was smaller than 15 mg/L. i.e. ideally one would compare the growth of *Arthrobacter* in the presence of DAA and to that in the absence of DAA. However, it is not clear if such a comparison would be feasible. Would it be possible to reduce the DAA concentration to a negligible level to conclusively test if *Arthrobacter* can obtain a benefit

from DAA? (e.g. using a different media?).

Response: The reviewer makes a good suggestion. To address this concern, we cultivated the *Arthrobacter* strains in M9 minimal medium with different DAA levels (Supplementary Fig. 19). The growth of *Arthrobacter* strains was inhibited by DAA at concentrations above 25 mg/L, compared to that in the absence of DAA. Therefore, it suggests that *Arthrobacter* strains did not benefit from DAA hydrolysis. We have revised the sentence as follows (**P15, L306-309**):

“Although the growth in ISEM and M9 minimal medium with different DAA levels suggests that *Arthrobacter* doesn’t gain fitness benefits through DAA hydrolysis (Supplementary Fig. 19), it receives return benefits from biofilm-forming species which secrete public goods.”

In the revised manuscript, the authors have also clarified how the biofilm and planktonic fractions were calculated, in addition to providing some additional details regarding the microfluidic experiments. However, a few questions about these experiments remain. In particular, the authors still have not addressed my previous comment about planktonic cells and the potential for contamination upstream and downstream of the porous media microfluidic device.

Response: In response to the comment, we have supplemented more details about the inoculation and sample collection. The colloidal transport and retention in the microfluidic chamber have been characterized to address the concerns about the retention of planktonic cells. We provided detailed responses to the specific comments raised by the reviewer below.

The device only has a single inlet and a single outlet. Were cells introduced into the device through the upstream tube? If so, then how did you prevent the tubing (whose volume is surely much larger than that of the device) from getting contaminated with bacteria? In practice, it is very difficult to avoid the contamination of tubing without switching tubes mid-experiment, which is technically difficult because it introduces air bubbles that detaches bacteria from surfaces. Can you please clarify how the devices were inoculated with bacteria and how the effluent was collected from the porous device without being “contaminated” by the tubing?

Response: Contamination of the upstream tubing would deplete nutrients and inhibit microbial growth in the microfluidic device. To avoid this issue, the bacterial cells were introduced to the microfluidic chamber via an inoculation port located downstream of the inoculation inlet (Supplementary Fig. 21). Regarding the effluent collection, we replaced the stainless steel needle and tubing at the outlet with sterile ones prior to sampling. Additional details have been added to the method section for clarification (**P19, L390-395 & P20, L396-403**).

“200 μL of soil bacterial suspension was introduced into microfluidic chips through an inoculation port located immediately downstream of the medium inlet (Supplementary Fig. 21A). The inlet tubing was clamped prior to inoculation to prevent backflow. The stainless steel needle was removed after injecting the inoculum followed by immediate sealing of the inoculation port with silicon glue (Supplementary Fig. 21B).”

“The stainless steel needle and tubing at the outlet were replaced with sterile ones before sampling. The effluent of microfluidic chips was collected to determine the cell density of planktonic communities. Thereafter, two sterile syringes were connected to the inlet and outlet of the microfluidic device. 1 mL of PBS buffer was repeatedly flushed back and forth between these two syringes at a flow rate of about 1,200 $\mu\text{L}/\text{min}$ for twenty times to extract all bacterial cells in the microfluidic chamber. The amount of planktonic and the total microbial cells in the microfluidic chips were quantified by qPCR using the primer pair Eub338F/Eub518R.”

In the revised methods, it is noted that the volume of the device is 3 μL , whilst the flow rate used in the experiment is 0.5 μL per minute, indicating that the residence time of planktonic bacteria within the device is only 6 mins. How then should one interpret the population dynamics of the “planktonic” bacteria, whose doubling time is surely larger than 6 mins? e.g. are the results being influenced by what is happening in the tubing upstream/downstream of the porous media device? e.g. are the changes in the numbers of planktonic *Arthrobacter* cells strongly influenced by changes in their rate of detachment from surfaces of the microfluidic device, rather than by changes in their growth rate? This is not clear and should be addressed.

Response: The reviewer brings up a very important point about the residence of planktonic cells in the microfluidic chamber. To address this concern, we characterized the transport and retention of fluorescent microspheres in the porous medium. Based on the analysis of the breakthrough curve, the mean travel time of microspheres in the chamber was more than 50 minutes. Previous studies have shown that the colloid-grain collisions contribute to an extended residence time compared to the theoretically expected residence time, especially in porous media with fine grain size^{7, 8}. It suggests that planktonic cells have sufficient residence time to grow in the microfluidic chamber. Moreover, we have prevented the contaminations from the tubing during inoculation and sample collection. Therefore, the changes in the amount of planktonic *Arthrobacter* were attributed to the combined effect of microbial growth in the ambient fluid and the continuous elution from the microfluidic chamber. In response to the reviewer’s comment, we have added the following sentences in the text (P6, L92-96 & P20, L415-426).

“The colloidal transport and retention in the microfluidic chamber were characterized to evaluate the residence of planktonic cells. Based on the breakthrough curve of fluorescent microspheres

(Supplementary Fig. 2), the mean travel time of colloid particles was 51.48 ± 2.16 minutes. It suggests that both biofilm and planktonic cells have sufficient residence time to grow in the porous medium.”

“Colloid transport and retention in the microfluidic chamber. To evaluate the residence time of planktonic cells in microfluidic chamber, we characterized the transport and retention of colloidal microspheres in the porous medium. The red fluorescent polystyrene microspheres purchased from Jiangsu Zhichuan Technology Co. (China) have a diameter of $2 \mu\text{m}$ and a density of 1.05 g/cm^3 . $3 \mu\text{L}$ of PBS buffer containing the fluorescent microspheres (1.0×10^5 particles/ μL) were pumped into the microfluidic chamber at a flow rate of $0.5 \mu\text{L}/\text{min}$. $1.5 \mu\text{L}$ of effluent was collected every 3 minutes at the outlet using a pipette and then diluted in $50 \mu\text{L}$ of PBS buffer in 384-well microplates. The particle density in the effluent was determined by measuring fluorescent intensity (excitation, 370 nm ; emission, 610 nm) (Supplementary Fig. 2B). Three independent replicates were carried out. The mean travel time (τ) was calculated as follows ^{9, 10}:

$$\tau = \frac{\int_0^T tC(t)dt}{\int_0^T C(t)dt}$$

where T is the duration of experiments, $C(t)$ is the particle density in the effluent at time point t.”

REFERENCE

7. Ausland G, Stevik TK, Hanssen JF, Köhler JC, Jenssen PD. Intermittent filtration of wastewater—removal of fecal coliforms and fecal streptococci. *Water Res.* **36**, 3507-3516 (2002).
8. Stevik TK, Ausland G, Hanssen JF, Jenssen PD. The influence of physical and chemical factors on the transport of *E. coli* through biological filters for wastewater purification. *Water Res.* **33**, 3701-3706 (1999).
9. Yu C, Warrick AW, Conklin MH. A moment method for analyzing breakthrough curves of step inputs. *Water Resour. Res.* **35**, 3567-3572 (1999).
10. Wu Y, Mohanty A, Chia WS, Cao B. Influence of 3-Chloroaniline on the Biofilm Lifestyle of *Comamonas testosteroni* and Its Implications on Bioaugmentation. *Appl. Environ. Microbiol.* **82**, 4401-4409 (2016).

Overall, I feel like this paper does constitute a significant advance. However, questions do remain as to whether (a) cooperation has been definitively demonstrated and (b) the interpretations of the population dynamics of planktonic bacteria within the (unavoidably small) microfluidic device that is presented are valid. If the former cannot be demonstrated definitively, then the claim that cooperation has been observed should be toned back significantly. In addition, if the reported phenomena is being strongly influenced by the population dynamics within the tubing on either side of the microfluidic device, then that should be spelled out more clearly.

Response: The increased cell numbers of *Arthrobacter* and biofilm-forming strains in co-culture mixtures indicated that the cooperative interactions improve the fitness of both biofilm and planktonic populations (Supplementary Fig. 5 & 6). We have included additional experimental results in the revised manuscript to demonstrate that the public goods excreted by biofilm-forming strains benefited *Arthrobacter*. Regarding the interpretations of planktonic bacteria, the colloidal breakthrough in the microfluidic chamber and more details about the experimental setup have been provided for clarification. We have addressed these concerns in the responses to the reviewer's comments above.

Minor comments:

Supplementary Fig. 10 - It is noted that the concentration of DAA in standard ISEM media is 44.1 mg/L. From the methods it appears that six different DAAs are included in the standard recipe for ISEM media. It would be informative to spell out how the different DAA concentrations reported in Supplementary Fig. 10 were prepared? e.g. do they contain all six different supplemented DAAs, but just at a lower concentrations than the standard media or ????. Please clarify.

Response: We prepared a concentrated DAA mixture containing equal amount of six DAAs. The concentrated stock solution was further diluted to prepare ISEM and M9 medium with different levels of DAAs. We have added the following sentences in the figure caption (Supplementary Fig. 19.).

“The relative growth of *Arthrobacter* strains in ISEM (A) and M9 minimal medium (B) supplemented with different levels of DAA after 48 h. A concentrated DAA stock solution with equal amounts of six DAAs (D-Val, D-Met, D-Leu, D-Phe, D-Thr and D-Trp) was diluted in ISEM and M9 medium to prepare different working concentrations.”

In Line 359 it says that “all bacterial cells including biofilms and planktonic cells were vigorously flushed out from the microfluidic chips with PBS buffer.” What do you mean by “vigorously”? e.g. what were the flow rates and timescales used? Please elaborate.

Response: To extract all bacterial cells from the microfluidic device, we connected two sterile syringes to the inlet and outlet and flushed PBS buffer between them at a flow rate of approximately 1,200 $\mu\text{L}/\text{min}$ for 20 cycles. We have added the following sentences to make it clearer (**P20, L397-400**).

“Thereafter, two sterile syringes were connected to the inlet and outlet of the microfluidic device. 1 mL of PBS buffer was repeatedly flushed back and forth between these two syringes at a flow rate of about 1,200 $\mu\text{L}/\text{min}$ for twenty times to extract all bacterial cells in the microfluidic chamber.”

Fig. 1D: The legend indicating which symbols corresponds to “biofilm” and which corresponds to “planktonic” have been removed in this version of the manuscript. These need to be added back in.

Response: We have added the figure legend in the revised manuscript.

Supplementary Fig. 15: The caption says that “Different letters indicate significant differences ($p < 0.05$, one-way ANOVA).”, but it does not state what “a” and “b” actually mean.

Response: Different letters in the figure represent significant differences in the relative growth of individual strains at different DAA levels. We have rephrased the sentence as follows (Supplementary Fig. 19):

“Different letters indicate significant differences in the relative growth of individual strains at different DAA levels ($p < 0.05$, one-way ANOVA), whilst shared letters indicate no statistical difference.”

Line 308-309: “micromolar” is repeated twice in this sentence. This passage needs to be reworded for clarity.

Response: We have reworded the sentence as follows (**P17, L337-339**):

“Micromoles of DAA per gram dry weight were detected in sediment and soil^{11, 12, 13}. As DAAs over this concentration range are sufficient to suppress biofilm formation^{14, 15}, they may therefore play a pervasive role in governing biofilm community structure over a wide range of ecosystems.”

REFERENCE

11. Langerhuus AT, *et al.* Endospore abundance and D: L-amino acid modeling of bacterial turnover in holocene marine sediment (Aarhus Bay). *Geochim. Cosmochim. Acta* **99**, 87-99 (2012).
12. Vranova V, *et al.* The significance of D-amino acids in soil, fate and utilization by microbes and plants: review and identification of knowledge gaps. *Plant Soil* **354**, 21-39 (2012).
13. Hu Y, Zheng Q, Zhang S, Noll L, Wanek W. Significant release and microbial utilization of amino sugars and D-amino acid enantiomers from microbial cell wall decomposition in soils. *Soil Biol. Biochem.* **123**, 115-125 (2018).
14. Kolodkin-Gal I, Romero D, Cao S, Clardy J, Kolter R, Losick R. D-amino acids trigger biofilm disassembly. *Science* **328**, 627-629 (2010).
15. Leiman SA, May JM, Lebar MD, Kahne D, Kolter R, Losick R. D-amino acids indirectly inhibit biofilm formation in *Bacillus subtilis* by interfering with protein synthesis. *J. Bacteriol.* **195**, 5391-5395 (2013).

Reviewer #3 (Remarks to the Author):

I thank the authors for adding additional conditioned medium growth experiments, siderophore measurements, and more method details during the revision.

The biggest concern about this work, as pointed out by all 3 reviewers during the last review, was that the “cooperation” focus of the manuscript was not supported. Now that by showing increased growth of *Arthrobacter* in the conditioned media of *Pseudomonas* and *Rhodococcus* and higher amounts of siderophore in the conditioned media, the authors concluded that *Pseudomonas* and *Rhodococcus* benefited *Arthrobacter* by producing siderophore as a public good. However, I don’t think the increased growth of *Arthrobacter* can be attributed to siderophore—which was not shown to directly improve growth here (esp. this was not a Fe-limiting condition) and siderophore is only one metabolite present in the complex mixture of conditioned media.

Response: We agree with the reviewer that the production of public goods does not necessarily imply that *Arthrobacter* can exploit them. To address this concern, we constructed a siderophore synthesis mutant. Deletion of the siderophore synthetase gene (*sfnaD*) in ASV1 *P. fluorescens* was found to reduce the growth of most *Arthrobacter* strains in the conditioned media (Supplementary Fig. 13). Siderophores have been shown to benefit bacteria in environments without iron limitation via transporting other essential nutrients, protecting from oxidative stress, and acting as signaling molecules¹⁶. For example, enterobactin and staphyloferrin have been reported to enhance microbial resistance to reactive oxygen species^{17, 18}. Moreover, we also found FAD and FMN secreted by ASV1 *P. fluorescens* enhanced the growth of *Arthrobacter* in both ISEM and M9 minimal medium (Supplementary Fig. 14). These results suggest that *Arthrobacter* can exploit the public goods excreted by the biofilm-forming strains. We have added the following sentences to make it clearer (P12, L230-241).

“On the other hand, ISEM conditioned by ASV1 *P. fluorescens* and ASV11 *R. erythropolis* significantly enhanced the growth of most *Arthrobacter* strains (Fig. 3D & Supplementary Fig. 10A). The enhanced growth was also observed in M9 minimal medium (Supplementary Fig. 10B), which suggested *Arthrobacter* strains benefited from the metabolites secreted by the biofilm-forming strains. As public goods such as siderophore, FMN and FAD were detected in the supernatant of ASV1 *P. fluorescens* grown in ISEM (Supplementary Fig. 11 & 12), we further investigated whether *Arthrobacter* exploited these public goods by constructing the siderophore synthesis mutant ($\Delta sfnaD$) and adding flavins to the culture media. Deletion of the siderophore synthetase gene reduced the growth of most *Arthrobacter* strains in the spent medium of ASV1 *P. fluorescens* (Supplementary Fig. 13). Meanwhile, the supplementation of flavins at the same level as the ISEM conditioned by ASV1 *P. fluorescens* significantly enhanced the growth of *Arthrobacter* in both

ISEM and M9 medium (Supplementary Fig. 14).”

REFERENCE

16. Johnstone TC, Nolan EM. Beyond iron: non-classical biological functions of bacterial siderophores. *Dalton Trans.* **44**, 6320-6339 (2015).
17. Nobre LS, Saraiva LM. Role of the siderophore transporter SirABC in the *Staphylococcus aureus* resistance to oxidative stress. *Curr. Microbiol.* **69**, 164-168 (2014).
18. Bogomolnaya L, Tilwawala R, Elfenbein J, Cirillo J, Andrews-Polymenis H. Linearized siderophore products secreted via MacAB efflux pump protect *Salmonella enterica* serovar Typhimurium from oxidative stress. *MBio* **11**, e00528-00520 (2020).

Furthermore, this still doesn't support the major conclusion that cooperative interaction mediated via extracellular metabolites caused the spatial segregation of free-living *Arthrobacter* and biofilm-forming bacteria. In particular, in addressing one previous comment, the authors added Time 0 inoculum community composition to Fig 2B, which shows *Pseudomonas* was the absolute dominant genus from the very beginning. So a plausible main mechanism could be as simple as that when the dominant *Pseudomonas* formed biofilms in ISEM (with ~44 mg/L DAA), the other dominant genus, biofilm-deficient *Arthrobacter*, was left behind with an increase in the relative abundance in the planktonic communities first, but then decreased to very low abundance under continuous flushing with fresh ISEM.

That said, I agree that the biofilm-promoting effects of *Arthrobacter*-degrading DAAs could still have contributed in situ, but likely to a less extent (<25%) considering the shorter time frame during early biofilm establishment (much <48h) and the overall decrease of DAAs (Fig 4D) and observed effects in pure culture (Fig 5B).

Response: The review raises a very important point regarding the prevalence of *Pseudomonas* in the microfluidic chamber. We agree that the high frequency of *Pseudomonas* in the inoculum contributed to their subsequent dominance during community succession. Remarkably, *Pseudomonas* transitioned its lifestyle from planktonic to biofilm in the early stage of biofilm development. *Pseudomonas* was the most dominant genus in the plankton at 12 h, while its relative abundance in the ambient fluid reduced significantly at 36 h. We analyzed the proportion of *Pseudomonas* in biofilm to the total *Pseudomonas* cell number in microfluidic chambers. The proportion of *Pseudomonas* inhabiting biofilm increased from 19.5±12.7% at 12 h to 97.3±10.0% at 36 h. The reduced frequency of planktonic *Pseudomonas* indicates the segregation of *Pseudomonas* from plankton. To validate whether the spatial segregation was induced via cooperative interaction, the proportions of *Pseudomonas* cells inhabiting biofilm in monoculture and co-culture containing *Arthrobacter* and *Pseudomonas* with comparable initial cell densities were compared. The proportions of *Pseudomonas* inhabiting biofilm to the total *Pseudomonas* cells

in co-culture were significantly higher than those in monoculture ($p < 0.001$, one-way ANOVA, Supplementary Table 1), which suggested that spatial segregation was induced by the cooperative interaction between free-living *Arthrobacter* and biofilm-forming bacteria. In response to this comment, we have added the following sentences in the text (P8, L146-150 & P15, L300-307).

“Based on qPCR and amplicon sequencing analyses (Fig. 1D, 2B & Supplementary Fig. 4), 19.5±12.7% of *Pseudomonas* in the microfluidic chamber inhabited biofilms at 12 h. The ratio further increased to 97.3±10.0% at 36 h. The increased ratio of sessile *Pseudomonas* cells suggests that *Pseudomonas* underwent a transition from planktonic to biofilm-forming in the early stage of biofilm development.”

“With the decreased level of biofilm inhibitors, *Pseudomonas*, the most abundant genus in the total community, reduced its proportion in the ambient fluid and switched to a sessile lifestyle. The spatial segregation was also observed in co-culture containing *Arthrobacter* and *Pseudomonas* strains with comparable initial cell densities. The pairwise interaction analyses revealed that *Arthrobacter* and biofilm-forming strains dominated the plankton and biofilm in the co-culture systems, respectively. The proportion of *Pseudomonas* that inhabited biofilms in co-culture was significantly higher than that in monoculture ($p < 0.001$, one-way ANOVA, Supplementary Table 1).”

Metabolomics: Even though it's level2 identification, the identification results should be provided, to include each metabolite's m/z, MS/MS similarity score, etc. Also, deposit of metabolomics spectra in MetaboLights under no. MTBLS3742 cannot be found.

Response: We have included m/z, retention time and MS/MS similarity score of each metabolite in the Source Data file. The raw data has also been deposited in the figshare database (https://figshare.com/projects/Cooperative_interactions_drive_spatial_segregation/156834).

Double check CLSM images in Source Data File; 72 h used a different stain compared with the others (Line94).

Response: We stained the microbial cells with DAPI (blue fluorescence) or SYTO 9 (green fluorescence) to quantify the biofilm growth. Both fluorescent dyes stain bacterial DNA and produce overlapping signals. The Pearson's correlation coefficient between these two fluorescent signals is 0.977±0.005 (Supplementary Fig. 24) (P21, L428-429).

“Both fluorescent dyes bind to genomic DNA and produce overlapping fluorescence (Supplementary Fig. 24).”

Line99: It's confusing to compare the cell densities of the planktonic vs biofilm communities. I can understand Equation in Line 370 at a single sampling timepoint, but it's confusing to think about it when considering the planktonic cells are continuously flushed out (and lost) during constant flow of fresh ISEM.

Also, "The amount of biofilm cells was considerably higher than that of the planktonic community" is not true in the early stage of incubation.

Response: To avoid confusion when comparing cell densities between planktonic and biofilm communities, we have calculated the amount of planktonic and biofilm cells in the microfluidic chamber to evaluate their growth. The following sentences have been revised (P20, L401-402, P20, L408-414 & P6, L106-107).

"The amount of planktonic and the total microbial cells in the microfluidic chips were quantified by qPCR using the primer pair Eub338F/Eub518R."

"The amount of biofilm cells was calculated as the difference between the total and planktonic cell numbers:

$$C_{planktonic} = D_{planktonic} \times V_{chip}$$

$$C_{biofilm} = C_{entire} - C_{planktonic}$$

where $C_{planktonic}$ and $C_{biofilm}$ are the planktonic and biofilm cell numbers in the microfluidic chamber, $D_{planktonic}$ is the cell density in the effluent of the microfluidic chamber, V_{chip} is the volume of the microfluidic chamber (3.0 μ L), and C_{entire} is the total cell number in the PBS eluate."

"Except for the initial 24 h, the amount of biofilm cells was considerably higher than that of the planktonic community."

Fig 1D: need to clarify which symbols represent planktonic vs biofilm.

Response: We have added the legend in the revised manuscript.

Line 173-185 and Supplementary Fig. 4-5 are a bit abrupt and confusing. Suggest to add some transition about how SFig4-5 were done and are related to Fig 3A-B, considering there are both monocultures and cocultures, and only contained part of the isolates in Fig3. Were SFig4-5 done in microfluidic devices? It's confusing why there are 3 different monoculture bars in the figures and how the inocula were constructed which seem to be of different compositions?

Response: The pairwise interaction analysis demonstrates that the interactions between *Arthrobacter* and biofilm-forming isolates, *Pseudomonas* and *Rhodococcus*, enhanced the overall

biofilm growth (Fig. 3A). Meanwhile, the biofilm biomass was measured via crystal violet assay. This method cannot infer the individual fitness of interaction partners in co-culture. Therefore, we quantified the abundance of *Arthrobacter* and biofilm-forming isolates under the same experimental conditions using qPCR (Supplementary Fig. 5 & 6). The bacterial culture was prepared and cultivated in microplates as shown in Fig. 3. The initial cell concentrations of the identical strain in the monoculture and co-culture were equal. The three monoculture bars in Supplementary Fig. 5 & 6 correspond to the planktonic, biofilm and total cell numbers. Following the reviewer's suggestion, we have added the following sentences to introduce the experimental setup for the pairwise interaction experiment (**P9, L172-173 & P10, L184-186 & P27, L563-565**).

“Their planktonic growth and biofilm formation in microplates were assayed by optical density measurements and crystal violet staining, respectively.”

“Since the crystal violet assay quantified the total biofilm biomass in co-culture, the abundance of *Arthrobacter* and biofilm-forming isolates in plankton, biofilm and total community was quantified using qPCR to evaluate their individual fitness.”

“The bacterial culture was prepared and cultivated as described above for the pairwise interaction analysis. The initial cell concentrations of the identical strain in the monoculture and different co-culture combinations were the same.”

Line 216: I thank the authors for adding the new spent medium experiments. This would need to revise Fig 3A and possibly add a subplot similar to 3D, to show the bidirectional growth promoting effects. Abstract needs to be revised to include this if keeping the current title.

Response: According to the reviewer's suggestion, we have added a new subplot for the result of conditioned medium in Fig. 3. The following sentence in abstract has been revised to include the new results (**P3, L43-47**):

“Through exometabolomics, transcriptomics, pairwise interaction analyses and genetic manipulation, we show that free-living *Arthrobacter* induces the surface colonization by scavenging the biofilm inhibitor, D-amino acids (DAAs) and receives benefits from the public goods secreted by the biofilm-forming strains.”

Line 278: Should add ISME without DAA addition as the control for comparison.

Response: As the soil extract contains approximately 15 mg/L of DAA, we assessed the growth of *Arthrobacter* strains in M9 minimal medium with and without DAA. The growth of *Arthrobacter*

strains was inhibited by DAA at concentrations above 25 mg/L, compared to that in the absence of DAA. Therefore, it suggests that *Arthrobacter* strains did not benefit from DAA hydrolysis. We have revised the sentence as follows (P15, L307-310):

“Although the growth in ISEM and M9 minimal medium with different DAA levels suggests that *Arthrobacter* doesn’t gain fitness benefits through DAA hydrolysis (Supplementary Fig. 19), it receives return benefits from biofilm-forming species which secrete public goods.”

Source data for Fig 3B, 3D need to be provided.

Response: We have provided the data for Figure 3 in the Source Data file (P41, L911-912).

“Source data for Fig. 3A and 3C are provided as a Source Data file. The measured OD₆₀₀ for Fig. 3D was plotted in Supplementary Fig. 10A.”

Line 416: Please clarify “Pooled QC samples were prepared by mixing an equal aliquot of effluent samples.” Also, please clarify what these QC are.

Response: We prepared the QC samples by pooling an equal volume of 42 effluent samples collected during the incubation. The following sentence has been revised for clarification (P23, L477-478).

“Quality control (QC) samples were prepared by pooling equal aliquots from all 42 effluent samples collected during the incubation period.”

Line 432: What is the MS/MS matching score?

Response: The cosine similarity score was used to indicate spectral similarity. This algorithm converts experimental and library spectra into two unit-vectors and determines their similarity based on the cosine of the angle between these two vectors. We have added the following sentence in the text to make it clearer (P24, L491-492).

“The MS/MS spectral similarity was represented using the cosine score¹⁹.”

REFERENCE

19. Stein SE, Scott DR. Optimization and testing of mass spectral library search algorithms for compound identification. *J. Am. Soc. Mass Spectrom.* **5**, 859-866 (1994).

Line 438: z-score may not be the ideal way here to analyze the data. Supplementary Fig. 12-14 may

be easier to understand if relative abundances are shown as changes compared with 0-h levels.

SFig13: It's hard to understand why cluster2 (consumed group) had metabolites positively correlated with time.

Response: Based on the reviewer's suggestion, we have reclassified all the metabolites based on Spearman's rank correlation between their dynamics and the incubation periods. Metabolites positively and negatively correlated with incubation period are included in clusters 1 and 2, respectively. The following sentences have been revised (**P24, L502-506**):

“All the exometabolites were grouped into three clusters based on Spearman's rank correlation between the incubation periods and the z score of each metabolite (Supplementary Fig. 15). Exometabolites with a correlation coefficient greater than 0.5 were considered as “released” and those with a coefficient less than -0.5 were considered as “consumed”²⁰. The remaining metabolites were categorized as “others”.”

REFERENCE

20. Swenson TL, Karaoz U, Swenson JM, Bowen BP, Northen TR. Linking soil biology and chemistry in biological soil crust using isolate exometabolomics. *Nat. Commun.* **9**, 19 (2018).

Line 472: I may have missed it somewhere, but couldn't find Supplementary Table 1.

Response: The method precision and accuracy were expressed as the relative standard deviation (RSD) and bias, respectively (Supplementary Table 2). We have revised the following sentences to make it clearer (**P26, L536-541**).

“QC samples were prepared at 1.0, 4.0 and 9.0 mg/L to assess the method precision and accuracy, which expressed as the relative standard deviation and bias, respectively (Supplementary Table 2).”

“The bias (%) was determined as (measured concentration - theoretical concentration)/theoretical concentration × 100.”

Line 496: Why this was done for 24 h but the other experiments, such as the conditioned medium experiment was done for 48 h.

Response: The microbial fitness was generally interpreted by the growth rate during exponential phase²¹. In our study, we quantified the growth of different strains at the exponential phase to infer their individual fitness in co-culture. Regarding the pairwise interaction and conditioned medium experiments, the culture was cultivated for 48 h to evaluate whether microbial interactions and

exometabolites influenced the overall biofilm formation and planktonic microbial growth at the stationary phase. We have revised the following sentence to make it clearer (P27, L564-566).

“After 24 h growth, the planktonic cells and biofilms attached on the inner surface of the microplates were collected at the exponential phase to evaluate the relative fitness of different strains²¹.”

REFERENCE

21. Hall BG, Acar H, Nandipati A, Barlow M. Growth Rates Made Easy. *Mol. Biol. Evol.* **31**, 232-238 (2014).

Reviewer comments, third round

Reviewer #1 (Remarks to the Author):

The authors have made a number of significant improvements to the manuscript which greatly strengthen their conclusions and have addressed my concerns. Cooperation in a true ecological and evolutionary sense (i.e a behaviour has evolved specifically to benefit its partner) is notoriously difficult to prove, but I believe the manuscript will spark discussion and make an impact in the field.

I have only a few minor comments:

To classify types of bacterial interactions the authors state:

'The co-culture interaction was defined as positive, when the biofilm yield of co-culture (Y_{co}) was higher than or equal to Y_{sum} ($Y_{co} \geq Y_{sum}$)'

Intuitively, $Y_{co} > Y_{sum}$ would indicate a positive interaction but $Y_{co} = Y_{sum}$ would indicate no interaction. The authors either need to change their definition or justify how $Y_{co} = Y_{sum}$ would also be a positive interaction.

Line 233: The authors should make clear that this is M9 media conditioned by *P. fluorescens* and *R. erythropolis*.

Line 237: The authors should clarify it is a siderophore synthesis mutant in *P. fluorescens*.

Supplementary Figure 26: I believe 'Nonsenses probe' is a typo.

Reviewer #2 (Remarks to the Author):

It is clear the author they have invested a great deal of time and effort developing new experiments to strengthen their conclusions.

I am pleased with the author responses to my previous comments and recommend publication.

Reviewer #3 (Remarks to the Author):

I thank the authors for making the efforts to address the additional comments, including adding experiments using siderophore synthesis mutant.

However, I'm still not quite persuaded by the thesis that cooperation caused spatial segregation of the free-living *Arthrobacter* and biofilm-forming bacteria. It could be that my lack of familiarity with microfluidic system is giving me a hard time to understand how *Arthrobacter* is benefiting in situ.

As commented previously, when the dominant *Pseudomonas* formed biofilms, the other dominant genus, biofilm-deficient *Arthrobacter*, was left behind with an increase in the relative abundance in the planktonic communities first, but then decreased to very low abundance under continuous flushing with fresh ISEM. I echo one other reviewer's concern on how population versus fluidic dynamics affects the interpretation of the benefit on planktonic populations in situ (not in spent medium or in 2 species co-culture experiments).

Point-by-Point Response to Reviewer's Comments and Criticisms

REVIEWERS' COMMENTS

Reviewer #1 (Remarks to the Author):

The authors have made a number of significant improvements to the manuscript which greatly strengthen their conclusions and have addressed my concerns. Cooperation in a true ecological and evolutionary sense (i.e a behaviour has evolved specifically to benefit its partner) is notoriously difficult to prove, but I believe the manuscript will spark discussion and make an impact in the field.

I have only a few minor comments:

To classify types of bacterial interactions the authors state:

‘The co-culture interaction was defined as positive, when the biofilm yield of co-culture (Y_{co}) was higher than or equal to Y_{sum} ($Y_{co} \geq Y_{sum}$)’

Intuitively, $Y_{co} > Y_{sum}$ would indicate a positive interaction but $Y_{co} = Y_{sum}$ would indicate no interaction. The authors either need to change their definition or justify how $Y_{co} = Y_{sum}$ would also be a positive interaction.

Response: We have revised the definition according to the reviewer's suggestion (P29, L587-591).

“The co-culture interaction was defined as positive, when the biofilm yield of co-culture (Y_{co}) was higher than Y_{sum} ($Y_{co} > Y_{sum}$). The relationship was considered as negative if the co-culture biofilm was less than or equal to the sum of the monocultures ($Y_{sum} \geq Y_{co}$). The co-culture interaction was strong negative when Y_{co} was less than Y_{min} , while a weak negative relationship was determined when $Y_{sum} \geq Y_{co} \geq Y_{min}$.”

Line 233: The authors should make clear that this is M9 media conditioned by *P. fluorescens* and *R. erythropolis*.

Response: The sentence has been rephrased as follows (P12, L228-230):

“The enhanced growth was also observed in M9 minimal medium conditioned by ASV1 *P. fluorescens* and ASV11 *R. erythropolis* (Supplementary Fig. 10b), which suggested *Arthrobacter* strains benefited from the metabolites secreted by the biofilm-forming strains.”

Line 237: The authors should clarify it is a siderophore synthesis mutant in *P. fluorescens*.

Response: We have reworded the following sentence for clarification (P12, L230-234).

“As public goods such as siderophore, FMN and FAD were detected in the supernatant of ASV1 *P. fluorescens* grown in ISEM (Supplementary Fig. 11 & 12), we further investigated whether *Arthrobacter* exploited these public goods by constructing the siderophore synthesis mutant in ASV1 *P. fluorescens* ($\Delta sfnaD$) and adding flavins to the culture media.”

Supplementary Figure 26: I believe ‘Nonsenses probe’ is a typo.

Response: The typo has been corrected.

Reviewer #2 (Remarks to the Author):

It is clear the author they have invested a great deal of time and effort developing new experiments to strengthen their conclusions.

I am pleased with the author responses to my previous comments and recommend publication.

Reviewer #3 (Remarks to the Author):

I thank the authors for making the efforts to address the additional comments, including adding experiments using siderophore synthesis mutant.

However, I’m still not quite persuaded by the thesis that cooperation caused spatial segregation of the free-living *Arthrobacter* and biofilm-forming bacteria. It could be that my lack of familiarity with microfluidic system is giving me a hard time to understand how *Arthrobacter* is benefiting in situ.

As commented previously, when the dominant *Pseudomonas* formed biofilms, the other dominant genus, biofilm-deficient *Arthrobacter*, was left behind with an increase in the relative abundance in the planktonic communities first, but then decreased to very low abundance under continuous flushing with fresh ISEM. I echo one other reviewer’s concern on how population versus fluidic dynamics affects the interpretation of the benefit on planktonic populations in situ (not in spent medium or in 2 species co-culture experiments).

Response: To address the reviewer’s concern, we compared the mean travel time of colloid particles

in the microfluidic chamber with and without biofilms (Supplementary Fig. 2). The result showed a significant decrease in the mean travel time from 51.48 ± 2.16 to 30.72 ± 5.16 minutes after the biofilm development (two-tailed Student's *t*-test, $n = 3$ independent replicates for each condition, $p = 0.011$). The reduced residence time was attributed to the decreased pore space resulting from biofilm formation^{1, 2}. Therefore, while the fitness of *Arthrobacter* was enhanced within the microbial community, the development of mature biofilm accelerated the elution of *Arthrobacter* from the microfluidic chips, ultimately leading to its extinction in the late stage. In response to the concern, we have added the following sentence for calcification (**P17, L330-333**).

“The mean travel time of colloids in the microfluidic chamber decreased from 51.48 ± 2.16 to 30.72 ± 5.16 minutes after the biofilm development (two-tailed Student's *t*-test, $n = 3$ independent replicates for each condition, $p = 0.011$, Supplementary Fig. 2), indicating an accelerated elution of planktonic *Arthrobacter* at the late stage.”

REFERENCE

1. Carrel M, et al. Biofilms in 3D porous media: Delineating the influence of the pore network geometry, flow and mass transfer on biofilm development. *Water Res.* 134, 280-291 (2018).
2. Coyte KZ, Tabuteau H, Gaffney EA, Foster KR, Durham WM. Microbial competition in porous environments can select against rapid biofilm growth. *Proc. Natl. Acad. Sci. U.S.A.* 114, E161-E170 (2017).